# Generalization and Knowledge Transfer in Abstract Visual Reasoning Models

## Abstract

We study generalization and knowledge reuse capabilities of deep neural networks in the domain of abstract visual reasoning (AVR), employing Raven's Progressive Matrices (RPMs), a recognized benchmark task for assessing AVR abilities. Two knowledge transfer scenarios referring to the I-RAVEN dataset are investigated. Firstly, inspired by generalization assessment capabilities of the PGM dataset and popularity of I-RAVEN, we introduce *Attributeless-I-RAVEN*, a benchmark with 10 generalization regimes that allow to test generalization of abstract rules applied to held-out attributes. Secondly, we construct *I-RAVEN-Mesh*, a dataset that enriches RPMs with a novel component structure comprising line-based patterns, facilitating assessment of progressive knowledge acquisition in transfer learning setting. The developed benchmarks reveal shortcomings of the contemporary deep learning models, which we partly address with *Pathways of Normalized Group Convolution (PoNG)* model, a novel neural architecture for solving AVR tasks. PoNG excels in both presented challenges, as well as the standard I-RAVEN and PGM setups. Encouraged by these promising results, we further evaluate PoNG in another AVR task, visual analogy problem with both synthetic and real-world images, demonstrating its strength beyond PRMs.

## 1 Introduction

Generalization, the ability of a model to perform well on unseen data, remains a fundamental challenge in deep learning (DL). While DL methods have demonstrated remarkable achievements in various domains, their generalization capabilities are often questioned, particularly in tasks that demand abstract problem-solving and reasoning skills (Chollet, 2019). One such domain is abstract visual reasoning (AVR) (Mitchell, 2021; van der Maas et al., 2021; Stabinger et al., 2021; Małkiński & Mańdziuk, 2023) that encompasses tasks requiring (human) fluid intelligence – an aspect of human cognition believed to be crucial for reasoning in never-encountered settings (Snow et al., 1984; Carpenter et al., 1990). The most popular AVR tasks are Raven's Progressive Matrices (RPMs) (Raven, 1936; Raven & Court, 1998), which constitute a common problem found in human IQ tests. Typical RPMs comprise two components – the context panels arranged in a $3 \times 3$ grid with the bottom-right panel missing and up to 8 answer panels, out of which only one correctly completes the matrix. Solving an RPM instance requires identification of underlying abstract rules applied to certain attributes of the objects composing the instance (see Fig. 1 for an illustrative example).

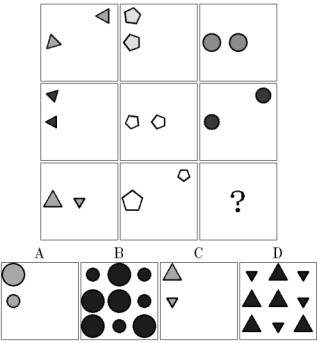

Figure 1: **RPM example.** The correct answer is A.

Design of computational methods capable of tackling RPMs has for decades been an active area of research (Evans, 1964; Gentner, 1980; Foundalis, 2006; Lovett et al., 2007; Kunda et al., 2010; Strannegård et al., 2013). Consequently, a number of works considered automatic creation of RPM datasets (Matzen et al., 2010; Wang & Su, 2015; Mańdziuk & Żychowski, 2019) and a wide suite of predictive models (Hernández-Orallo et al., 2016; Hernández-Orallo, 2017) were proposed, with DL methods showing the most promising performance (Yang et al., 2022; Małkiński & Mańdziuk, 2022). While this rapid progress led to exceeding the human level in particular problem setups (Wu

et al., 2020; Mondal et al., 2023), a fundamental challenge of generalization to novel problem settings remains largely unattained.

Initial works designed several RPM datasets (Matzen et al., 2010; Wang & Su, 2015; Hoshen & Werman, 2017), however, measuring generalization was not their focus. While some works explored knowledge transfer between related tasks (Mańdziuk & Żychowski, 2019; Tomaszewska et al., 2022), the complexity of the datasets was limited and consequently they didn't pose a challenge for contemporary DL methods. To measure generalization in modern DL models, the PGM dataset was introduced (Barrett et al., 2018). PGM defines eight generalization regimes, each specifying the distribution of objects, rules and attributes in train and test splits. For instance, in the `Held-out Triples` split, a given rule–object–attribute triplet (e.g. Progression on Object's Size) was assigned only to one of the two splits. In effect, the models were tested on triplet combinations different from training ones, allowing to assess their generalization capabilities. A subsequent work proposed RAVEN (Zhang et al., 2019a), another RPM dataset with enriched perceptual complexity of matrices instantiated in seven visual configurations (`Center`, `2x2Grid`, `3x3Grid`, `Left-Right`, `Up-Down`, `Out-InCenter`, `Out-InGrid`). Moreover, the benchmark is characterized by a moderate sample size, i.e. 70K instances, compared to 1.42M RPMs per each of the eight regimes in PGM. Due to this size disparity, subsequent research gravitated towards RAVEN and its revised variants (I-RAVEN (Hu et al., 2021) and RAVEN-Fair (Benny et al., 2021)), which didn't require substantial computational resources to train DL models.

Drawing inspiration from the broad adoption of RAVEN and the generalization assessment capabilities of PGM, this paper proposes a novel suite of generalization challenges stemming from I-RAVEN (Hu et al., 2021) (a revised variant of RAVEN that removes a bias in RAVEN's answer panels). However, unlike I-RAVEN, the proposed suite of benchmarks allows for a direct assessment of the generalization and knowledge transfer of AVR models. Compared to PGM, our datasets feature compositionality and variety of figure configurations, and their processing doesn't require substantial computational resources. Furthermore, they include structural annotations, which are utilized, for example, in recent neuro-symbolic approaches.

First, we introduce *Attributeless-I-RAVEN*, comprising 10 generalization regimes. The 4 primary regimes correspond to specific held-out attributes ({`Position`, `Type`, `Size`, `Color`}), resp. The training matrices in these regimes adhere to the `Constant` rule for the respective attribute, whereas test matrices employ a rule different from `Constant` for this attribute (i.e., `Progression`, `Arithmetic`, or `Distribute Three`). Moreover, we propose 6 extended regimes: 3 of them feature a held-out attribute pair, while another 3 replace the `Constant` rule in the training set with each remaining rule. In effect, each regime comprises different distributions of training and test data.

Next, we propose *I-RAVEN-Mesh*, a variant of I-RAVEN with a new grid-like structure overlaid on the matrices. The dataset enables assessing generalization to incrementally added structures and progressive knowledge acquisition in a transfer learning (TL) setting.

In investigations involving 13 contemporary AVR DL models, we observed that the introduced benchmarks present a substantial challenge for the tested methods. This prompted the development of *Pathways of Normalized Group Convolution (PoNG)*, a novel AVR model that excels in both problem setups: generalization to held-out attributes and incremental knowledge acquisition.

Our main contributions can be summarized as follows:

- We introduce the *Attributeless-I-RAVEN* (A-I-RAVEN) dataset that enables measuring generalization across 10 regimes.

- We construct *I-RAVEN-Mesh*, an extension of I-RAVEN with a new component structure that facilitates assessment of progressive knowledge acquisition in a TL setting.

- We evaluate the performance of state-of-the-art AVR models on the introduced benchmarks, uncovering their limitations in terms of generalization to novel problem settings.

- We propose a new neural architecture for solving AVR tasks termed PoNG, which excels in addressing both introduced challenges, as well as the standard I-RAVEN and PGM setups. Additionally, PoNG demonstrates the state-of-the-art performance in visual analogy problem (VAP) in both synthetic and real-world setups.

## 2 RELATED WORK

**Generalization in AVR.** In recent years, a variety of AVR problems and corresponding datasets have emerged (Bongard, 1968; Nie et al., 2020; Fleuret et al., 2011; Qi et al., 2021; Shanahan et al., 2020; Jiang et al., 2024; Hill et al., 2019; Zhang et al., 2020) and several attempts have been made to measure generalization in contemporary AVR models based on the introduced benchmarks. In particular, distinct visual configurations were employed in RAVEN to assess how a model trained on one configuration performs on the remaining ones (Zhang et al., 2019a; Spratley et al., 2020; Zhuo & Kankanhalli, 2021). Although in such a setting the visual aspects of train/test matrices come from different distributions, the underlying rules and attributes remain the same. In contrast, A-I-RAVEN enables studying the generalization of rules applied to held-out attributes, shifting the focus from perception towards reasoning. Besides RPMs, the limits of generalization have been explored in other AVR tasks as well. Visual Analogy Extrapolation Challenge evaluates model's capacity for extrapolation (Webb et al., 2020). However, such specialized datasets might favor models that explicitly embed the notion of extrapolation in their design and aim for being invariant only to specific attributes such as object size or location. Differently, our benchmarks allow verifying the model's capacity to learn a given concept from the data and generalize it to novel settings. This perspective links our work to the recent literature on concept learning (Odouard & Mitchell, 2022; Moskvichev et al., 2023). However, the concept-oriented benchmarks that originate from ARC (Chollet, 2019) remain largely unsolved by DL models and pose a significant challenge even for leading multi-modal large language models (Mitchell et al., 2023). In contrast, both benchmarks proposed in this work are attainable by DL models, though further advances in generalization abilities of the models are necessary to consider them solved.

**Model architectures.** Preliminary attempts to solve RPMs with DL models involve WReN (Barrett et al., 2018) that reasons over object relations using Relation Network (Santoro et al., 2017), or SRAN (Hu et al., 2021) that relies on a hierarchical architecture with panel encoders devoted to particular image groups. A common theme enabling generalization in DL models is to explicitly identify RPM objects. To this end, RelBase (Spratley et al., 2020) employs Attend-Infer-Repeat (Eslami et al., 2016), an unsupervised scene decomposition method, STSN (Mondal et al., 2023) utilizes Slot attention (Locatello et al., 2020) to decompose matrix to slots containing particular objects and Temporal Context Normalization (TCN) (Webb et al., 2020) to normalize latent matrix panel representations in a task-specific context, DRNet (Zhao et al., 2024) relies on a dual-stream design, and MRNet (Benny et al., 2021) presents a multi-scale architecture. SCL (Wu et al., 2020) proposes the scattering transformation, CoPINet (Zhang et al., 2019b) and CPCNet (Yang et al., 2023b) rely on contrastive architectures, PredRNet (Yang et al., 2023a) learns to minimize the prediction error, ALANS (Zhang et al., 2021) and PrAE (Zhang et al., 2022a) employ neuro-symbolic architectures, and SCAR (Małkiński & Mańdziuk, 2024b) adapts its computation to the structure of the considered matrix. Despite the high variety of AVR models, experiments on the introduced benchmarks reveal their shortcomings in terms of generalization and knowledge transfer. In this context, we propose PoNG, a new AVR model that excels in the presented tasks by combining parallel architecture, weight sharing, and tactical normalization.

## 3 METHODS

The set of attributes in I-RAVEN is $\mathcal{A} = \{\texttt{Position}, \texttt{Number}, \texttt{Type}, \texttt{Size}, \texttt{Color}\}$ and the set of rules is $\mathcal{R} = \{\texttt{Constant}, \texttt{Progression}, \texttt{Arithmetic}, \texttt{Distribute Three}\}$. For attribute $a \in \mathcal{A}$ and a dataset split $s \in \mathcal{S}$, where $\mathcal{S} = \{\text{train, val., test}\}$, we define the set of rules applicable to $a$ in split $s$ by $R(a, s) \subseteq \mathcal{R}$. In I-RAVEN all rule–attribute pairs are valid in all splits:

$$R(a, s) = \mathcal{R}, \quad \forall a \in \mathcal{A} \wedge \forall s \in \mathcal{S} \tag{1}$$

### 3.1 ATTRIBUTELESS-I-RAVEN

To probe generalization in DL models, we present A-I-RAVEN, a benchmark composed of 10 generalization regimes. Example matrices are illustrated in Fig. 2, with additional samples provided in Appendix A. Each regime defines a set of held-out attributes $A^*$, each with a corresponding rule $r^*(a), a \in A^*$. In train and validation splits, held-out attribute $a \in A^*$ is governed by $r^*(a)$. In

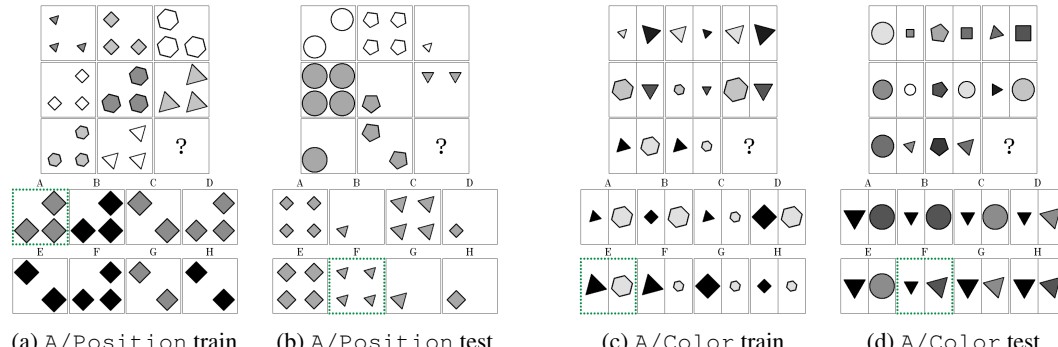

(a) `A/Position` train     (b) `A/Position` test     (c) `A/Color` train     (d) `A/Color` test

Figure 2: **Attributeless-I-RAVEN.** Left: Matrices from the `A/Position` regime belonging to the `2×2 Grid` configuration. In (a), object position is constant across rows, while in (b) object numerosity is governed by `Distribute Three`. Right: Matrices from the `A/Color` regime belonging to the `Left-Right` configuration. In (c), object color is constant across rows in left and right image parts, while in (d) it's governed by `Progression`. Correct answers are marked in a green dotted border. Please refer to Appendix A for examples from other generalization regimes.

the test split, $a \in A^*$ is governed by a different rule sampled from $\mathcal{R} - \{r^*(a)\}$. In effect, during training, the model doesn't see rule–attribute combinations required to solve test matrices. There are no rule-related constraints on the remaining attributes. In summary, we have:

$$R(a,s) = \begin{cases} \{r^*(a)\} & \text{if } a \in A^* \wedge s \in \{\text{train}, \text{validation}\}, \\ \mathcal{R} - \{r^*(a)\} & \text{if } a \in A^* \wedge s = \text{test}, \\ \mathcal{R} & \text{if } a \notin A^*. \end{cases} \tag{2}$$

We define 4 primary regimes with $r^*(a) = $ `Constant` that correspond to individual held-out attributes ($|A^*| = 1$), denoted as `A/<Attribute>` (e.g., `A/Type`). Since `Position` and `Number` attributes are tightly coupled (e.g., it's impossible to increase cardinality of objects while keeping their position constant), we allocate a single generalization regime, `A/Position`, to cover both attributes. In addition, we define 6 extended regimes as supplementary generalization challenges. In the first group a pair of attributes is held-out in the training set, i.e. $|A^*| = 2$. Specifically, we introduce 3 new regimes: `A/ColorSize`, `A/ColorType`, and `A/SizeType`, based on the respective attribute pairs. In the second group, `Constant` rule in $r^*(a)$ is replaced with each of the 3 remaining rules, leading to `A/Color-Progression`, `A/Color-Arithmetic`, and `A/Color-DistributeThree` regimes. While this modification could be applied to all the described regimes, we focus on the `Color` attribute due to its broad range of possible values.

### 3.2   I-RAVEN-MESH

The other of the proposed benchmarks is designed to probe progressive knowledge acquisition in a TL setting. I-RAVEN-Mesh extends I-RAVEN by introducing a novel visual component overlaid on top of the existing I-RAVEN components (see Fig. 3). Though the dataset can serve as a learning challenge on its own, the main motivation behind its introduction is to employ models pre-trained on I-RAVEN and fine-tune them on I-RAVEN-Mesh with a configurable train sample size, facilitating analysis of their TL performance. The mesh grid comprises from 1 to 12 lines placed in predefined locations. The set of available lines covers the inner and outer edges of a $2 \times 2$ grid (12 lines in total). The mesh component has two attributes: $\mathcal{A}^{\text{mesh}} = \{\text{Number}, \text{Position}\}$, which govern the count and location of lines, respectively. To each attribute a rule $r \in \mathcal{R}$ can be applied. Table 1 describes the effect of applying a given rule–attribute pair to the mesh component.     To generate the mesh component of an I-RAVEN-Mesh matrix, we sample one of the two attributes $a \in \mathcal{A}^{\text{mesh}}$ and a corresponding rule $r \in \mathcal{R}$ that governs its values. As the attributes often depend on each other (e.g., it's impossible to increase the number of lines while keeping their position constant), we don't constrain the value of the other attribute. The rule–attribute pairs for the base I-RAVEN components are generated in the same way as in the original dataset. To generate answers to the matrix, we follow the impartial algorithm proposed in I-RAVEN (Hu et al., 2021). In addition, each matrix contains

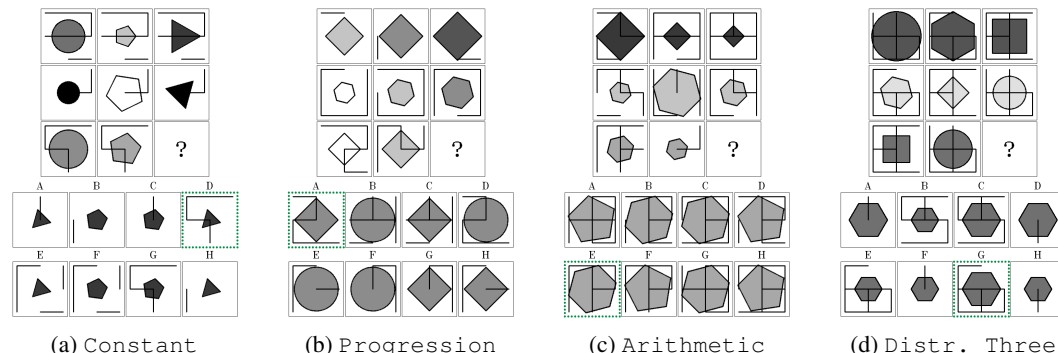

(a) `Constant`      (b) `Progression`      (c) `Arithmetic`      (d) `Distr. Three`

Figure 3: **I-RAVEN-Mesh.** Matrices with the `Position` attribute of the mesh component governed by all applicable rules. For the sake of readability, we present examples belonging to the `Center` configuration. (a) Line position is constant in each row. (b) The line pattern displayed in the first column is rotated by 90 degrees in subsequent columns. (c) The union set operator applied to the first and the second column produces line positions in the third column. (d) Each row contains lines arranged in one out of three available patterns. Correct answers are marked in a green dotted border. Please refer to Appendix A for examples concerning the `Number` attribute.

Table 1: Description of rule–attribute pairs in I-RAVEN-Mesh. D3 marks `Distribute Three`.

| Attribute | Rule | Description |
|---|---|---|
| Number | Constant | Each image in a given row contains the same number of lines. |
| | Progression | The count of lines in a given row changes by a constant factor (e.g. $2, 4, 6$). |
| | Arithmetic | The number of lines in the third column is determined based on an arithmetic operation applied to the preceding columns (e.g. $3 - 1 = 2$). |
| | D3 | Three line counts are sampled and spread among images in a given row. |
| Position | Constant | Each image in a given row contains the same position of lines. |
| | Progression | A panel arrangement is sampled in each row and rotated by 90 degrees in subsequent columns. |
| | Arithmetic | The position of lines in the third column is computed based on a set operation (union or difference) applied to the preceding columns. |
| | D3 | Three line arrangements are sampled and spread among images in a given row. |

at least one incorrect answer that differs from the correct one only in the mesh component, ensuring that the solver has to identify the correct rule governing the mesh component in order to solve the matrix. To facilitate training with an auxiliary loss, in which the model additionally predicts the representation of rules governing the matrix (Barrett et al., 2018), we extend the base set of rule annotations with ones concerning the Mesh component.

### 3.3 PATHWAYS OF NORMALIZED GROUP CONVOLUTION (PONG)

In initial experiments, we've found out that SOTA AVR models struggle in the proposed generalization challenges. Consequently, we introduce PoNG (Fig. 4), a novel model that outcompetes baselines across a number of problem settings. The model follows a typical two-stage design. Firstly, it generates an embedding of each image panel. Then, it aggregates representations of matrix panels to predict the index of the correct answer. The details are described in Appendix D.

Let $(X, y, r)$ denote an RPM, where $X = \{x_i\}_{i=1}^{16}$ is the set of image panels comprising 8 context panels $\{x_i\}_{i=1}^{8}$ and 8 answer panels $\{x_i\}_{i=9}^{16}$, $x_i \in [0, 1]^{h \times w}, i = 1, \ldots, 16$ is a grayscale image of height $h$ and width $w$, $y \in \{0, 1\}^8$ is the one-hot encoded index of the correct answer, $r \in \{0, 1\}^{d_r}$ is the multi-hot encoded representation of matrix rules of dimensionality $d_r$ using sparse encoding (Małkiński & Mańdziuk, 2024a). In each experiment $h = w = 80$, while $d_r$ is determined by the number of matrix components in the corresponding dataset ($d_r = 48$ for I-RAVEN-Mesh, $d_r = 40$ otherwise; see Appendix C for details).

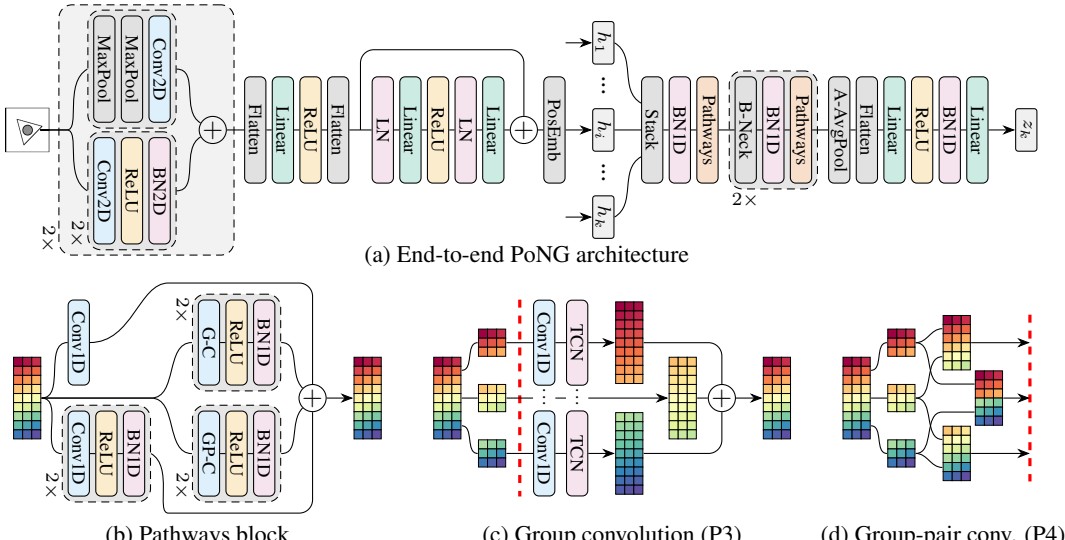

(a) End-to-end PoNG architecture

(b) Pathways block  (c) Group convolution (P3)  (d) Group-pair conv. (P4)

Figure 4: **PoNG.** (a) The panel encoder embeds each input image $x_i$ independently, producing $h_i$. Context panel embeddings $\{h_i\}_{i=1}^8$ together with the embedding of $k$'th answer $h_k$ are stacked and processed with the reasoner, leading to $z_k$. (b) The pathways block, a key component of PoNG, comprises four parallel pathways P1 – P4. (c) P3 and (d) P4 employ novel normalized group convolution operators. PosEmb denotes position embedding, G-C the group convolution module used in P3, and GP-C the group-pair convolution module used in P4. The red dashed line marks the point after which G-C and GP-C perform analogous computation.

**Panel encoder.** The first component of the model has the form $\mathcal{E} : x \to h$, where $h \in \mathbb{R}^{d_h}$ is the input panel embedding of dimensionality $d_h$. Following RelBase (Spratley et al., 2020), the module comprises 2 blocks of the same architecture. Each block includes 2 parallel pathways that build high-level and low-level features, resp. The first one contains 2 convolutional blocks, each with 2D convolution, ReLU, and Batch Normalization (BN) (Ioffe & Szegedy, 2015). The second one contains 2D max pooling followed by 2D convolution. The sum of both pathway results forms the block output. Differently from RelBase, we flatten the height and width dimensions of the resultant embedding, pass it through a linear layer with ReLU, flatten the channel and spatial dimensions, and pass the tensor through a feed-forward residual block with Layer Normalization (LN) (Ba et al., 2016). Finally, we concatenate the tensor with a position embedding (a learned 25-dimensional vector for each cell in the $3 \times 3$ context grid), leading to $h$.

**Reasoner.** The second component of the model has the form $\mathcal{R} : \{h_i\}_{i=1}^8 \cup h_k \to z_k$, where $h_k$ is the panel embedding of $k$'th answer. For each answer panel, the reasoner produces embedding $z_k$ that describes how well the considered answer fits into the matrix context. Panel embeddings $\{h_i\}_{i=1}^8 \cup h_k$ are stacked and processed by a sequence of 3 reasoning blocks interleaved with 2 bottleneck layers for dimensionality reduction. Each reasoning block comprises BN and 4 parallel pathways, outputs of which are added together to form the output of the block. Next, the latent representation is passed through adaptive average pooling, flattened, processed with a linear layer with ReLU, passed through BN and projected with a linear layer to $z_k \in \mathbb{R}^{128}$.

**Pathways.** The key aspect of the reasoner module are its pathways. Each takes an input tensor of shape $(B, C, D)$, where $B$ is the batch size, $C$ is the number of channels, and $D$ is the feature dimension. In the first reasoning block $D = d_h$ and $C = 9$ corresponds to the number of panel embeddings in the considered group. Pathways are described as follows: P1 – a pointwise 1D convolution layer that mixes panel features at each spatial location; P2 – a sequence of 2 blocks, each comprising 1D convolution, ReLU, and BN, that builds higher level features spanning neighbouring spatial locations; P3 – analogous to P2, but 1D convolution is replaced with a group 1D convolution that splits the tensor into several groups along the channel dimension, applies a 1D convolution with shared weights to each group, and adds together the representations of each group; P4 – analogous

to P3, but groups are arranged into pairs concatenated along the channel dimension and processed with a 1D convolution with shared weights. In contrast to (Krizhevsky et al., 2012), the proposed group convolution layers in both P3 and P4 apply TCN (Webb et al., 2020) to the outputs in each group. In the first layer P3 and P4 split the input tensor into 3 groups, which allows for producing embeddings of each matrix row and each pair of rows, resp. Though we apply the pathways block in the RPM context, we envisage it as a generic module, also applicable to other settings involving a set of vector representations of shape $(B, C, D)$.

**Answer prediction.** Eight representations of the context matrix filled-in with the respective answer, $\{z_k\}_{k=1}^8$, are processed with three prediction heads. The target head $\mathcal{P}^y : z_k \rightarrow \widehat{y_k}$ employs two linear layers interleaved with ReLU to produce score $\widehat{y_k} \in \mathbb{R}$ describing how well the answer $k$ aligns with the matrix context. The aggregate rule head $\mathcal{P}_1^r : \{z_k\}_{k=1}^8 \rightarrow \widehat{r_1}$ computes the sum of inputs and processes it with two linear layers interleaved with ReLU, producing a latent prediction of matrix rules $\widehat{r_1} \in \mathbb{R}^{d_r}$. The target-conditioned rule head $\mathcal{P}_2^r : \{z_k\}_{k=1}^8 \rightarrow \widehat{r_2}$ processes its input through a linear layer and computes a weighted sum of the resultant embeddings with weights given by the predicted probability distribution over the set of possible answers $\sigma(\{\widehat{y_k}\}_{k=1}^8)$, where $\sigma$ denotes softmax. The model is trained with a joint loss function $\mathcal{L} = \text{CE}(\sigma(\{\widehat{y_k}\}_{k=1}^8), y) + \beta\text{BCE}(\zeta(\widehat{r_1}, r)) + \gamma\text{BCE}(\zeta(\widehat{r_2}, r))$, where $\zeta$ denotes sigmoid, CE cross-entropy, BCE binary cross-entropy, $\beta = 25$ and $\gamma = 5$ are balancing coefficients.

## 4 EXPERIMENTS

We assess generalization of state-of-the-art models for solving RPMs on A-I-RAVEN, evaluate progressive knowledge acquisition on I-RAVEN-Mesh, and conduct an ablation study to showcase the contribution of the respective modules that constitute PoNG. We also evaluate PoNG on two additional VAPs comprising synthetic (Hill et al., 2019) and real-world (Bitton et al., 2023) images.

**Experimental setup.** In all experiments we use the Adam optimizer (Kingma & Ba, 2014) with $\beta_1 = 0.9, \beta_2 = 0.999, \epsilon = 10^{-8}$ and a batch size set to 128. Learning rate is initialized to 0.001 and reduced 10-fold (at most 3 times) if no progress is seen in the validation loss in 5 subsequent epochs, and training stops early in the case of 10 epochs without progress. Unless stated otherwise, each model configuration was trained 3 times with a different seed, and we report mean and standard deviation for these runs. In each experiment, we utilize 42 000 training, 14 000 validation, and 14 000 test matrices, following the standard data split protocol taken in prior works (Zhang et al., 2019a; Hu et al., 2021). All reference models are trained with the auxiliary loss with sparse encoding and $\beta = 1$. Experiments were run on a worker with a single NVIDIA DGX A100 GPU.

**Baselines.** In addition to the simple CNN-LSTM baseline (Barrett et al., 2018), we assess generalization of SOTA AVR models including WReN (Barrett et al., 2018), CoPINet (Zhang et al., 2019b), RelBase (Spratley et al., 2020), SCL (Wu et al., 2020), MRNet (Benny et al., 2021), ALANS (Zhang et al., 2021), SRAN (Hu et al., 2021), PrAE (Zhang et al., 2022a), CPCNet (Yang et al., 2023b), PredRNet (Yang et al., 2023a), STSN (Mondal et al., 2023), and DRNet (Zhao et al., 2024). For direct comparison, we evaluate all models on I-RAVEN following the above-described experimental setup.

**Reproducibility.** To guarantee reproducibility of experiments, we use a fixed set of random seeds and turn off hardware and framework features concerning indeterministic computation wherever possible. Together with the code, we provide the full training script that can be used to run all training jobs. The training job is packaged as a Docker image with fixed dependencies to isolate the configuration of the training environment. The released code allows for generation of all datasets from scratch, eliminating the dependency on file-hosting services required to distribute the data. The code for reproducing all experiments is publicly accessible at: <hidden-for-blind-review>.

**Generalization on Attributeless-I-RAVEN.** In the first set of experiments we evaluate all considered models on 4 primary generalization regimes of A-I-RAVEN. The results are presented in Table 2, along with the reference results on I-RAVEN and I-RAVEN-Mesh. PoNG outperforms all selected baselines across all settings. Among baseline models, the best results on A/Color and A/Position are achieved by DRNet, followed by RelBase and SCL. In the remaining attributeless regimes, SCL outperforms other baselines with DRNet taking the second place. Interestingly,

Table 2: **Single-task learning.** Mean and standard deviation of test accuracy for three random seeds. Best dataset results are marked in bold and the second best are underlined. Pos. denotes Position. I-RVN[†] denotes results on I-RAVEN reported by model authors in the corresponding papers.

| | I-RVN[†] | I-RAVEN | Mesh | A/Color | A/Pos. | A/Size | A/Type |
|---|---|---|---|---|---|---|---|
| ALANS | – | 27.0 ($\pm$ 8.4) | 15.9 ($\pm$ 2.6) | 15.2 ($\pm$ 1.4) | 16.0 ($\pm$ 1.0) | 23.3 ($\pm$ 6.5) | 19.0 ($\pm$ 3.4) |
| CPCNet | **98.5** | 70.4 ($\pm$ 6.4) | 66.6 ($\pm$ 5.1) | 51.2 ($\pm$ 3.8) | 68.3 ($\pm$ 4.0) | 43.5 ($\pm$ 3.5) | 38.6 ($\pm$ 4.3) |
| CNN-LSTM | 18.9 | 27.5 ($\pm$ 1.5) | 28.9 ($\pm$ 0.4) | 17.0 ($\pm$ 3.1) | 24.0 ($\pm$ 2.9) | 13.6 ($\pm$ 1.4) | 14.5 ($\pm$ 0.8) |
| CoPINet | 46.1 | 43.2 ($\pm$ 0.1) | 41.1 ($\pm$ 0.3) | 32.5 ($\pm$ 0.2) | 41.3 ($\pm$ 1.6) | 21.8 ($\pm$ 0.2) | 19.8 ($\pm$ 0.9) |
| DRNet | 97.6 | 90.9 ($\pm$ 1.1) | 83.9 ($\pm$ 2.7) | 70.0 ($\pm$ 1.6) | 77.5 ($\pm$ 0.9) | 54.3 ($\pm$ 3.0) | 44.3 ($\pm$ 0.8) |
| MRNet | 83.5 | 86.7 ($\pm$ 2.3) | 79.5 ($\pm$ 2.0) | 33.6 ($\pm$ 8.2) | 62.6 ($\pm$ 2.6) | 20.6 ($\pm$ 5.0) | 19.4 ($\pm$ 0.3) |
| PrAE | 77.0 | 19.5 ($\pm$ 0.4) | 33.2 ($\pm$ 0.4) | 47.9 ($\pm$ 0.9) | 68.2 ($\pm$ 3.3) | 41.3 ($\pm$ 1.8) | 37.0 ($\pm$ 1.7) |
| PredRNet | 96.5 | 88.8 ($\pm$ 1.8) | 59.2 ($\pm$ 6.4) | 59.4 ($\pm$ 1.0) | 73.7 ($\pm$ 0.7) | 47.5 ($\pm$ 1.3) | 40.2 ($\pm$ 1.3) |
| RelBase | 91.1 | 89.6 ($\pm$ 0.6) | 84.9 ($\pm$ 4.4) | 67.4 ($\pm$ 2.7) | 76.6 ($\pm$ 0.3) | 51.1 ($\pm$ 2.4) | 44.1 ($\pm$ 1.0) |
| SCL | 95.0 | 83.4 ($\pm$ 2.5) | 80.9 ($\pm$ 1.5) | 65.1 ($\pm$ 2.0) | 76.7 ($\pm$ 7.1) | 65.6 ($\pm$ 2.4) | 49.5 ($\pm$ 1.8) |
| SRAN | 60.8 | 58.2 ($\pm$ 1.6) | 57.8 ($\pm$ 0.2) | 38.3 ($\pm$ 1.0) | 56.9 ($\pm$ 0.7) | 34.4 ($\pm$ 3.0) | 30.7 ($\pm$ 2.2) |
| STSN | 95.7 | 51.0 ($\pm$ 24.8) | 48.7 ($\pm$ 11.5) | 39.3 ($\pm$ 6.9) | 36.1 ($\pm$ 19.9) | 38.4 ($\pm$ 16.6) | 39.1 ($\pm$ 5.0) |
| WReN | 23.8 | 18.4 ($\pm$ 0.0) | 25.7 ($\pm$ 0.2) | 16.9 ($\pm$ 0.5) | 17.3 ($\pm$ 0.4) | 12.4 ($\pm$ 0.5) | 15.1 ($\pm$ 0.7) |
| PoNG (ours) | 95.9 | **95.9** ($\pm$ 0.7) | **89.3** ($\pm$ 2.4) | **80.3** ($\pm$ 4.3) | **79.3** ($\pm$ 0.7) | **73.5** ($\pm$ 3.1) | **59.4** ($\pm$ 6.9) |

Table 3: **PoNG ablations**. Test accuracy averaged across 3 random seeds and a difference to the default model setup (cf. Table 2). Union denotes application of all ablations but the first one.

| | I-RAVEN | Mesh | A/Color | A/Pos. | A/Size | A/Type |
|---|---|---|---|---|---|---|
| w/o P1 and P2 | 92.8 (− 3.1) | 74.4 (−14.9) | 73.3 (− 7.0) | 76.4 (− 2.9) | 58.4 (−15.2) | 49.5 (− 9.8) |
| w/o P3 and P4 | 95.6 (− 0.3) | 88.0 (− 1.3) | 78.9 (− 1.4) | 78.6 (− 0.7) | 73.9 (+ 0.4) | 53.9 (− 5.5) |
| w/o TCN | 96.0 (+ 0.1) | 90.8 (+ 1.4) | 75.4 (− 4.9) | 80.3 (+ 1.0) | 66.6 (− 6.9) | 57.5 (− 1.9) |
| $\gamma = 0$ | 95.7 (− 0.1) | 88.8 (− 0.5) | 74.2 (− 6.1) | 79.6 (+ 0.3) | 73.0 (− 0.5) | 56.9 (− 2.5) |
| $\beta = 0$ | 94.2 (− 1.7) | 91.4 (+ 2.1) | 79.0 (− 1.3) | 77.5 (− 1.8) | 70.3 (− 3.2) | 53.3 (− 6.1) |
| $\gamma = 0 \wedge \beta = 0$ | 79.7 (−16.2) | 32.7 (−56.7) | 72.1 (− 8.2) | 75.1 (− 4.2) | 64.9 (− 8.6) | 49.0 (−10.3) |
| union | 81.4 (−14.5) | 32.5 (−56.8) | 76.2 (− 4.1) | 74.1 (− 5.2) | 66.9 (− 6.6) | 46.0 (−13.4) |

the top 3 models rely on rather shallow architectures, yet outcompete other methods that rely on a deeper layout, such as SRAN or STSN. This suggests that parameter-efficient AVR models not only excel in solving RPMs but also generalize better.

Generalization regimes of A-I-RAVEN pose a bigger challenge than the base dataset. While PoNG, the best performing model, achieved 95.9% test accuracy on I-RAVEN, on A-I-RAVEN regimes it scored from 59.4% (on A/Type) to 80.3% (on A/Color). Fig. 5 displays the difference in PoNG's performance on test and validation splits. On I-RAVEN and I-RAVEN-Mesh the difference is negligible, as in these datasets both splits follow the same distribution. However, the difference in attributeless regimes is significant, which indicates the need for further research on generalization. In Appendix E we present further evaluation on 6 extended A-I-RAVEN regimes. As shown in Table 10, replacing the Constant rule in the training set with Progression or DistributeThree yields

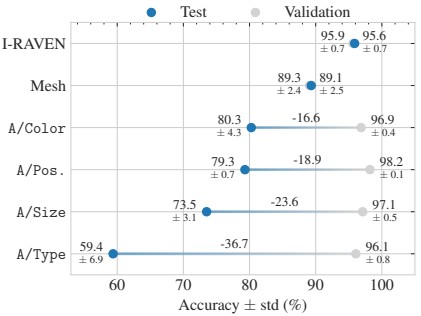

Figure 5: **Dataset difficulty.** PoNG's performance on test and validation splits.

a dataset of similar complexity (the best model achieves 81.4 / 81.3% accuracy), while using the Arithmetic rule increases the difficulty (the best model scored 70%). Furthermore, using a pair of held-out attributes significantly increases the complexity. For instance, in A/SizeType, the most challenging regime, the best result is only 33.5%. Notably, PoNG outperforms all other models in 5 out of 6 settings. We conclude that A-I-RAVEN provides a suite of challenging regimes of variable complexity, in which even the best performing models are far from solving all test matrices.

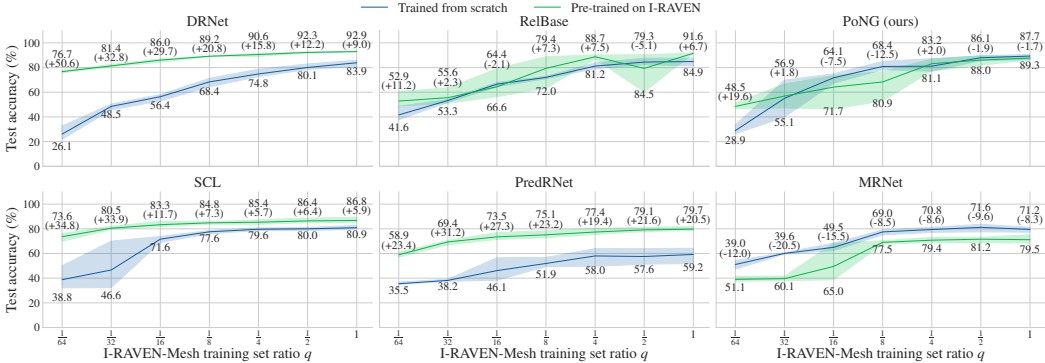

Figure 6: **Transfer learning.** Mean and standard deviation of test accuracy on I-RAVEN-Mesh across three random seeds. Models were trained in two setups: 1) from scratch on I-RAVEN-Mesh with variable sample size; 2) pre-trained on full I-RAVEN and fine-tuned on I-RAVEN-Mesh with variable sample size. Results for setups 1) and 2) are shown below and above the plot lines, resp.

Table 4: **PGM**. Test accuracy of PoNG in all regimes of the PGM dataset. The Interpolation regime is denoted as Inter., Held-out Attribute Pairs as HO-AP, Held-out Triple Pairs as HO-TP, Held-out Triples as HO-Triples, Held-out Attribute line-type as HO-LT, Held-out Attribute shape-colour as HO-SC, and Extrapolation as Extra. For reference, we provide results of SCL (Wu et al., 2020; Małkiński & Mańdziuk, 2024a), MRNet (Benny et al., 2021), ARII (Zhang et al., 2022b), PredRNet (Yang et al., 2023a), DRNet Zhao et al. (2024), and Slot-Abstractor (Mondal et al., 2024).

| Model | Neutral | Inter. | HO-AP | HO-TP | HO-Triples | HO-LT | HO-SC | Extra. | Avg. |
|---|---|---|---|---|---|---|---|---|---|
| SCL | 87.1 | 56.0 | 79.6 | 76.6 | 23.0 | 14.1 | 12.6 | 19.8 | 46.1 |
| MRNet | 93.4 | 68.1 | 38.4 | 55.3 | 25.9 | **30.1** | **16.9** | 19.2 | 43.4 |
| ARII | 88.0 | 57.8 | 50.0 | 64.1 | 32.1 | 16.0 | 12.7 | _29.0_ | 43.7 |
| PredRNet | 97.4 | 70.5 | 63.4 | 67.8 | 23.4 | 27.3 | 13.1 | 19.7 | 47.8 |
| DRNet | **99.1** | _83.8_ | **93.7** | 78.1 | **48.8** | _27.9_ | 13.1 | 22.2 | **58.3** |
| Slot-Abstractor | 91.5 | **91.6** | 63.3 | _78.3_ | 20.4 | 16.7 | _14.3_ | **39.3** | 51.9 |
| PoNG (ours) | _98.1_ | 75.2 | _92.1_ | **97.7** | _46.1_ | 16.9 | 12.6 | 19.9 | _57.3_ |

**Progressive knowledge acquisition on I-RAVEN-Mesh.** In the second set of experiments we employ I-RAVEN-Mesh to examine the TL ability of the best performing models (see Appendix E for extended results). To this end, we consider variants of partial I-RAVEN-Mesh dataset with a fraction $q \in \{\frac{1}{64}, \ldots, 1\}$ of the training set and compare the performance of a model trained from scratch on a partial dataset to that of a model pre-trained on full I-RAVEN and fine-tuned on a part of I-RAVEN-Mesh. Fig. 6 shows that for $q = \frac{1}{64}$ pre-training RelBase and MRNet on I-RAVEN leads to gains smaller than 15 p.p., whereas pre-training DRNet, PoNG, SCL and PredRNet improved their accuracy by 50.6, 19.6, 34.8 and 23.4 p.p., resp. In addition, TL clearly improved performance of DRNet, SCL and PredRNet in all considered settings, in particular for $q = 1$ by 9.0, 5.9 and 20.5 p.p., resp., indicating their capacity for knowledge reuse.

**Ablation study.** We performed an ablation study with simplified PoNG variants. Table 3 presents the results. The removal of P1 and P2 leads to performance drop, in particular on `I-RAVEN-Mesh` ($-14.9$ p.p.) and `A/Size` ($-15.2$ p.p.). Similarly, removing P3 and P4 reduces model performance, especially on `A/Type` ($-5.5$ p.p.). Disabling TCN leads to generally worse results, primarily on `A/Color` ($-4.9$ p.p.) and `A/Size` ($-6.9$ p.p.). Training without $\mathcal{P}_2^r$ ($\gamma = 0$) or $\mathcal{P}_1^r$ ($\beta = 0$) typically reduces model performance, but training with one of these rule-based prediction heads compensates to some degree the lack of the other. However, the removal of both ($\gamma = 0 \wedge \beta = 0$) deteriorates results across all datasets, signifying high relevance of the auxiliary training signal in PoNG's training. To confirm the inherent out-of-distribution generalization abilities of PoNG, we evaluated the model on all PGM regimes without performing any hyperparameter optimization (we only changed the batch size to 256 to reduce training time). Table 4 shows that PoNG achieves strong results on PGM, particularly on the Held-out Triple Pairs regime, exceeding the best reference model

Table 5: **Visual Analogy Problems (Hill et al., 2019).** Results of LBC, NSM, and PredRNet come from (Yang et al., 2023a, Table 2d). For PoNG, we present mean and std of test accuracy for three random seeds. ND denotes Novel Domain, NTD — Novel Target Domain, NAV — Novel Attribute Values, Inter. — Interpolation, Extra. — Extrapolation.

| | ND Transfer | NTD LineType | NTD ShapeColor | NAV Inter. | NAV Extra. | Avg |
|---|---|---|---|---|---|---|
| LBC | $0.87 \pm 0.005$ | $0.76 \pm 0.020$ | $0.78 \pm 0.004$ | $0.93 \pm 0.004$ | $0.62 \pm 0.020$ | 0.79 |
| NSM | 0.88 | 0.79 | 0.78 | 0.93 | **0.74** | 0.82 |
| PredRNet | $0.96 \pm 0.003$ | **0.82** $\pm 0.010$ | $0.80 \pm 0.010$ | $0.97 \pm 0.002$ | $0.72 \pm 0.060$ | **0.85** |
| PoNG (ours) | **0.98** $\pm 0.001$ | $0.78 \pm 0.006$ | **0.81** $\pm 0.006$ | **0.98** $\pm 0.000$ | $0.68 \pm 0.007$ | 0.84 |

Table 6: **VASR (Bitton et al., 2023).** Results of selected baselines come from (Bitton et al., 2023, Table 3). For PoNG, we present mean with std and best-of-3 test accuracy for three random seeds. Sup. denotes Supervised.

| Distractors | Zero-Shot ViT | Zero-Shot Swin | Sup. Concat | PoNG (best-of-3) | PoNG (mean $\pm$ std) |
|---|---|---|---|---|---|
| Random | 86.0 | 86.0 | 84.1 | **92.0** | $91.8 \pm 0.3$ |
| Difficult | 50.3 | 52.9 | 54.9 | **70.5** | $69.5 \pm 1.1$ |

by 19.4 p.p. We conclude that strong performance of PoNG on A-I-RAVEN and I-RAVEN-Mesh should not be attributed to any specific bias of the model towards these two datasets.

**Synthetic visual analogies.** The VAP benchmark (Hill et al., 2019) was introduced to assess the analogy-based reasoning capabilities of the learning systems. It comprises five generalization regimes: Novel Domain Transfer, Novel Target Domain: Colour of Shapes, Novel Target Domain: Type of Lines, Novel Attribute Values: Interpolation, Novel Attribute Values: Extrapolation, which test the model's generalization to novel domains or attribute values. We compared PoNG to results reported in (Yang et al., 2023a, Table 2d), a recent paper introducing the PredRNet model that achieves SOTA results across most VAP regimes. We run PoNG with three random seeds and present its average test accuracy and standard deviation. The results are showcased in Table 5. PoNG presents best results in 3 out of 5 settings, showing its applicability to AVR tasks beyond RPMs.

**Real-world visual analogies.** The VASR dataset (Bitton et al., 2023) presents visual analogies comprising real-world images. In effect, the learner needs to additionally understand a rich real-world scene, before attempting to solve the presented analogy problem. Following the approach proposed by the VASR authors, we employed the Vision Transformer (ViT) (Dosovitskiy et al., 2021) as a perception backbone that produces image embeddings. Specifically, we used the same model variant as (Bitton et al., 2023), which is ViT-L/32 pre-trained on ImageNet-21k at resolution 224x224 and fine-tuned on ImageNet-1k at resolution 384x384. We replaced the panel encoder of PoNG with this frozen pre-trained backbone and trained the rest of the model from scratch. We evaluated the model on two VASR splits including random and difficult distractors, resp. As shown in Table 6, in both cases our model outcompetes the strongest result among baselines with 92.0% vs. 86.0% and 70.5% vs. 54.9%, resp. The results support the claim that PoNG is a versatile model with strong analogical reasoning capabilities, applicable to both synthetic and real-world domains.

## 5 CONCLUSION

We investigate generalization capabilities of DL models in the AVR domain. To accelerate research in this area, we propose two RPM benchmarks. Attributeless-I-RAVEN introduces 10 generalization regimes of variable complexity that assess model's capability to solve matrices with rules applied to novel attributes. I-RAVEN-Mesh overlays line-based patterns on top of the RPM, facilitating TL studies. Experiments on 13 strong literature AVR models reveal their limitations in terms of generalization. To elevate state-of-the-art, we introduce PoNG, a novel AVR model capitalizing on parallel design, weight sharing, and normalization. PoNG outcompetes all baselines on the presented challenges, and achieves significant improvement over SOTA reference models on PGM. Furthermore, PoNG excels in solving visual analogy problems comprising both synthetic and real-world images.

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

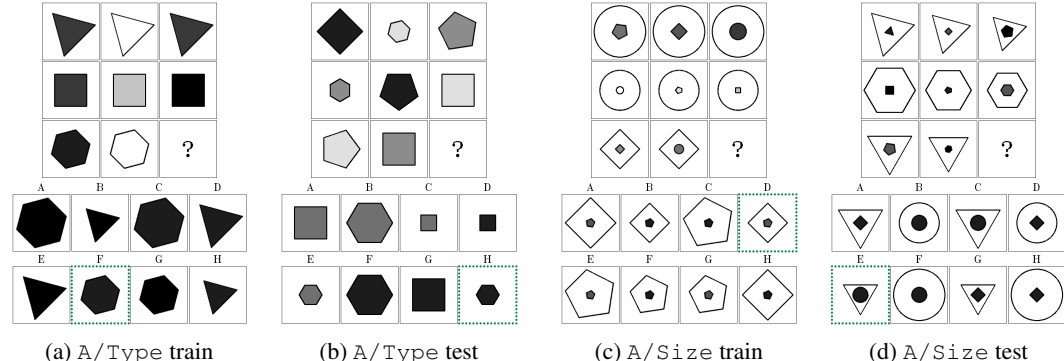

| (a) `A/Type` train | (b) `A/Type` test | (c) `A/Size` train | (d) `A/Size` test |
|---|---|---|---|

Figure 7: **Attributeless-I-RAVEN.** Left: Matrices from the `A/Type` regime belonging to the `Center` configuration. In (a), object type is constant across rows, while in (b) it's governed by the `Distribute Three` rule. Right: Matrices from the `A/Size` regime belonging to the `Out-InCenter` configuration. In (c), object size is constant across rows in both inner and outer image parts, while in (d) the inner and outer components are governed by the `Arithmetic` and `Progression` rules, resp.

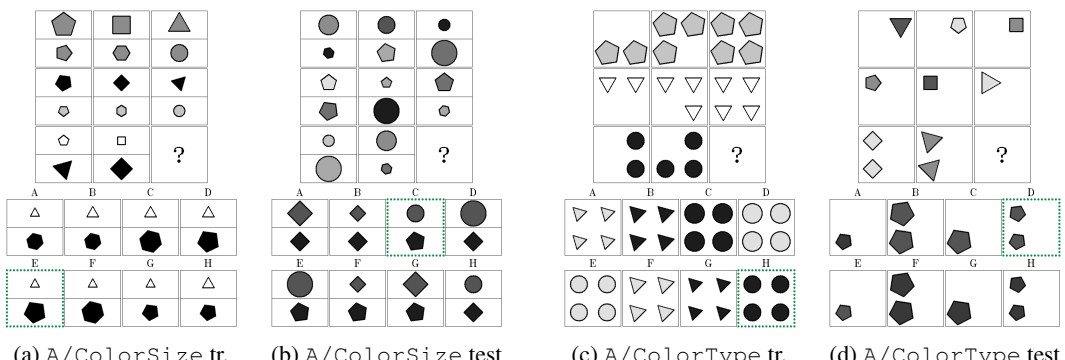

| (a) `A/ColorSize` tr. | (b) `A/ColorSize` test | (c) `A/ColorType` tr. | (d) `A/ColorType` test |
|---|---|---|---|

Figure 8: **Attributeless-I-RAVEN.** Left: Matrices from the `A/ColorSize` regime belonging to the `Up-Down` configuration. In (a), object color and size is constant across rows in both components, while in (b) they are governed by `Progression` and `Distribute Three` in the upper component, resp., and by `Distribute Three` in the lower one. Right: Matrices from the `A/ColorType` regime belonging to the `2x2 Grid` configuration. In (c), object color and type is constant across rows, while in (d) they are governed by the `Distribute Three` rule.

## A  ADDITIONAL MATRIX EXAMPLES

Figure 7 presents matrix examples from `A/Type` and `A/Size`, the primary regimes of Attributeless-I-RAVEN. Figures 8, 9 and 10 depict matrix examples from the extended regimes of Attributeless-I-RAVEN: `A/ColorSize` and `A/ColorType` (Fig. 8), `A/SizeType` and `A/Color-Progression` (Fig. 9), and `A/Color-Arithmetic` and `A/Color-DistributeThree` (Fig. 10). Figure 11 presents matrix examples from I-RAVEN-Mesh concerning the `Number` attribute.

## B  LIMITATIONS AND FUTURE WORK

In this work we study generalization and knowledge transfer in contemporary AVR models employing RPM datasets, and compare the introduced PoNG model with SOTA models in solving visual analogy problems. However, the set of problems in the AVR domain also includes other tasks not covered in the paper (Małkiński & Mańdziuk, 2023). The Machine Number Sense dataset presents visual arithmetic problems (Zhang et al., 2020), VAEC defines an extrapolation challenge (Webb

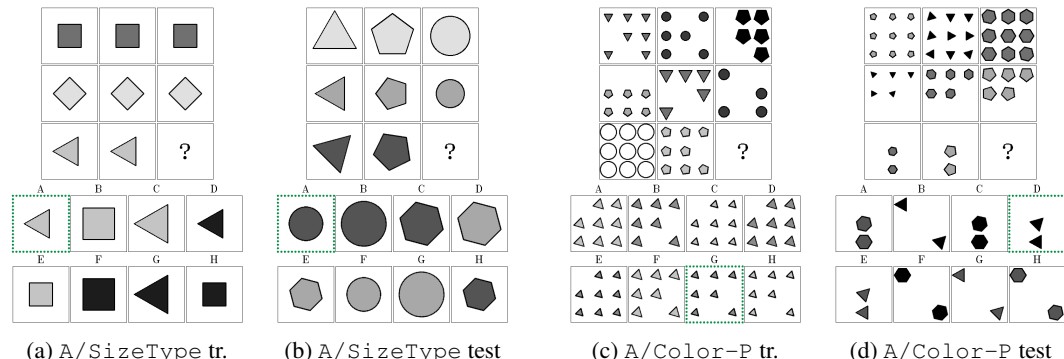

(a) A/SizeType tr.    (b) A/SizeType test    (c) A/Color-P tr.    (d) A/Color-P test

Figure 9: **Attributeless-I-RAVEN.** Left: Matrices from the A/SizeType regime belonging to the Center configuration. In (a), object size and type are constant across rows, while in (b) they are governed by the Progression rule. Right: Matrices from the A/Color-Progression regime belonging to the 3x3 Grid configuration. In (c), object color is governed by the Progression rule, while in (d) by Distribute Three.

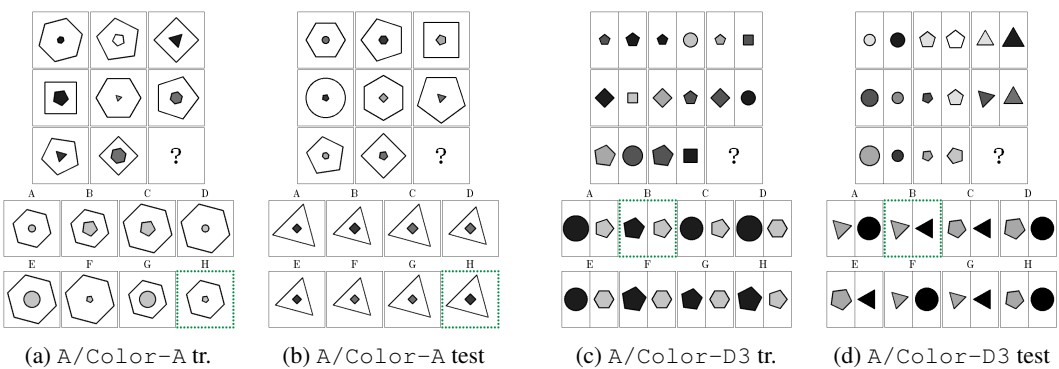

(a) A/Color-A tr.    (b) A/Color-A test    (c) A/Color-D3 tr.    (d) A/Color-D3 test

Figure 10: **Attributeless-I-RAVEN.** Left: Matrices from the A/Color-Arithmetic regime belonging to the Out-InCenter configuration. In (a), object color in the inner component is governed by Arithmetic, while in (b) it is governed by Distribute Three. Right: Matrices from the A/Color-DistributeThree regime belonging to the Left-Right configuration. In (c), object color is governed by the Distribute Three rule in both components, while in (d) by Constant in the left component and by Arithmetic in the right one.

et al., 2020), while ARC proposes a set of diverse tasks in a few-shot learning setting (Chollet, 2019). Future research in this area may juxtapose the performance of AVR models across a set of benchmarks oriented towards generalization to ensure generalization advances beyond the RPM and visual analogy datasets.

In the paper we claim that the proposed pathways block, a key component of the introduced model, is a generic module also applicable to other tasks that require reasoning over a set of objects (vector embeddings). Nevertheless, the experimental evaluation of PoNG presented in the paper is focused on RPM benchmarks, including I-RAVEN, I-RAVEN-Mesh, Attributeless-I-RAVEN, and PGM, and two visual analogy datasets, i.e. VAP and VASR. Assessing model's performance on other problems constitutes an interesting extension of this work.

## C DATASET DETAILS

**Rule encoding.** As discussed in Section 3.3, we use sparse encoding (Małkiński & Mańdziuk, 2024a) to represent the set of matrix rules as a vector $r \in \mathbb{R}^{d_r}$, such that $d_r = 48$ for I-RAVEN-Mesh and $d_r = 40$ otherwise. The set of rules $\mathcal{R}$ in I-RAVEN is {Constant, Progression, Arithmetic, Distribute Three} and the set of attributes $\mathcal{A}$ is {Position, Number,

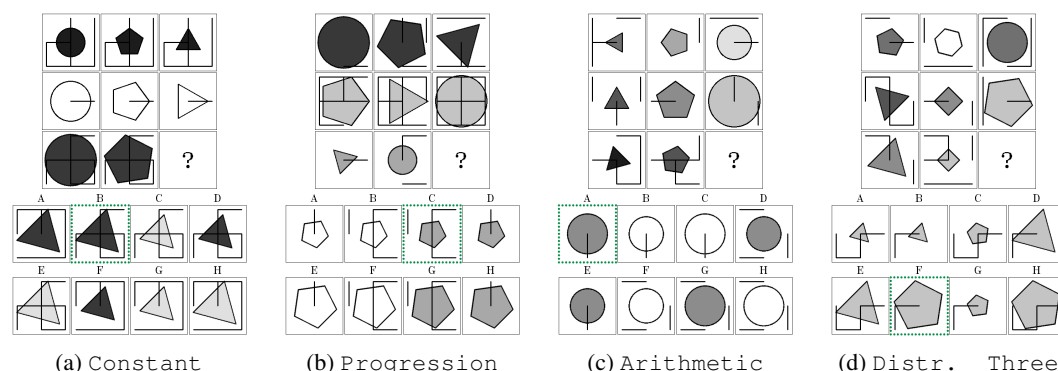

| (a) Constant | (b) Progression | (c) Arithmetic | (d) Distr. Three |

Figure 11: **I-RAVEN-Mesh.** The examples showcase matrices with the Number attribute of the mesh component governed by all applicable rules. (a) Line number is constant in each row. (b) The number of lines increases by 2 from left to right. (c) The number of lines in the third column is the difference between the number of lines in the second and first columns. (d) The numbers of lines in each row compose a set $\{2, 3, 6\}$.

Type, Size, Color}. It follows that there is $|\mathcal{R}| \times |\mathcal{A}| = 20$ unique rule–attribute pairs. In addition, the Left-Right, Up-Down, Out-InCenter, and Out-InGrid configurations in I-RAVEN comprise two components in which rules exist independently, e.g., the Left-Right component contains matrices with separate rules applied to the left and right sides. This gives an upper bound of 40 rule–attribute combinations in each configuration. As discussed in Section 3.2 and presented in Table 1, the Mesh component introduced in I-RAVEN-Mesh comprises two attributes and four rules, leading to a total of 48 rule–attribute combinations per configuration. As an example, in the Up-Down configuration of I-RAVEN-Mesh, there are 20 rule–attribute combinations for the upper component, another 20 for the lower component, and 8 for the Mesh component. The sparse encoding encodes each rule in a matrix as a one-hot vector and applies the OR operation to the set of one-hot vectors, producing a multi-hot representation of matrix rules.

## D MODEL DETAILS

Tables 7, 8, and 9 list all PoNG hyperparameters.

## E EXTENDED RESULTS

Fig. 12 presents extended results of Fig. 6.

Table 10 shows the aggregated performance of all considered models on 6 extended A-I-RAVEN regimes.

Tables 11 – 22 present the results (mean and standard deviation) of all considered models on test and validation splits and the difference between these two splits for particular datasets/regimes. The results support the analysis of dataset difficulty presented in Section 4. The difference in model performance between test and validation splits in I-RAVEN (Table 11) and I-RAVEN-Mesh (Table 12) is negligible. In Attributeless-I-RAVEN regimes, however, the difference is significant, showing limitations of all evaluated models in terms of generalization. Across 4 primary regimes (Tables 13 – 16), the biggest difference concerns the A/Type regime, suggesting that generalization of rules applied to novel shape types constitutes a real challenge for the contemporary models. In all 3 extended regimes concerning held-out attribute pairs (A/ColorSize, A/ColorType, and A/SizeType) the performance difference on test and validation splits is bigger than in the primary regimes (see Tables 17 – 19). This drop stems from overall weaker performance on the test split, confirming high difficulty of these regimes. Model performance on the next 3 regimes concerning the Color attribute and rules other than Constant (A/Color-Progression, A/Color-Arithmetic, and A/Color-DistributeThree) is better, though further progress in generalization is required to fully close the performance gap between test and validation splits (see Tables 20 – 22).

Table 7: **PoNG hyperparameters: Panel encoder** $\mathcal{E}$**.** The parameters of convolution layers are denoted as [# input channels $\rightarrow$ # output channels, kernel size, stride, padding]; of pooling layers as [kernel size, stride, padding]; of linear layers as [# input neurons $\rightarrow$ # output neurons]; of flatten operators as [input dimensions $\rightarrow$ output dimensions]; of position embedding as [dimensionality of the position embedding vector].

| LAYER | HYPERPARAMETERS |
|---|---|
| CONV2D-RELU-BN2D | $[1 \rightarrow 32, 7 \times 7, 2 \times 2, 3 \times 3]$ |
| CONV2D-RELU-BN2D | $[32 \rightarrow 32, 7 \times 7, 2 \times 2, 3 \times 3]$ |
| MAXPOOL | $[3 \times 3, 2 \times 2, 1 \times 1]$ |
| MAXPOOL | $[3 \times 3, 2 \times 2, 1 \times 1]$ |
| CONV2D | $[1 \rightarrow 32, 1 \times 1, 1 \times 1, 0 \times 0]$ |
| SUM | |
| CONV2D-RELU-BN2D | $[32 \rightarrow 32, 7 \times 7, 2 \times 2, 3 \times 3]$ |
| CONV2D-RELU-BN2D | $[32 \rightarrow 32, 7 \times 7, 2 \times 2, 3 \times 3]$ |
| MAXPOOL | $[3 \times 3, 2 \times 2, 1 \times 1]$ |
| MAXPOOL | $[3 \times 3, 2 \times 2, 1 \times 1]$ |
| CONV2D | $[32 \rightarrow 32, 1 \times 1, 1 \times 1, 0 \times 0]$ |
| SUM | |
| FLATTEN (HEIGHT & WIDTH) | $[5 \times 5 \rightarrow 25]$ |
| LINEAR-RELU | $[25 \rightarrow 25]$ |
| FLATTEN (DEPTH & SPATIAL) | $[32 \times 25 \rightarrow 800]$ |
| LN | |
| LINEAR-RELU | $[800 \rightarrow 1600]$ |
| LN | |
| LINEAR | $[1600 \rightarrow 800]$ |
| SUM | |
| POSITION EMBEDDING | $[25]$ |

Tables 23 – 34 present the results (mean and standard deviation) of all considered models in detail for all matrix configurations. The most challenging configurations in I-RAVEN and I-RAVEN-Mesh are `3x3Grid` and `Out-InGrid`, in which image panels contain more objects than in the remaining configurations. Apparently, such setups require stronger reasoning capabilities to correctly identify the rules applied to multiple objects. Also, the results on the `Left-Right` and `Up-Down` configurations are relatively weaker in majority of regimes. In these configurations, rules may be applied to both matrix components (left/right and up/down, resp.), which demands stronger reasoning capabilities. This also concerns the `Out-InGrid` configuration in the `A/Size` regime, and the `Out-InCenter` configuration in the `A/SizeType` regime. Results in the `A/Position` regime are close-to-perfect in configurations comprising a single object in each component (`Center`, `Left-Right`, `Up-Down`, and `Out-InCenter`) and weaker in the remaining configurations (`2x2Grid`, `3x3Grid` and `Out-InGrid`). This performance drop can be attributed to the fact that `Position` attribute can only be effectively applied to the `2x2Grid`, `3x3Grid` and `Out-InGrid` configurations allowing modification of the object's position. In the remaining configurations its application does not introduce any changes.

Table 8: **PoNG hyperparameters: Reasoner $\mathcal{R}$.** The parameters of convolution layers are denoted as [# input channels $\rightarrow$ # output channels, kernel size, stride, padding]; of group and group-pair convolution layers as [# input channels $\rightarrow$ # output channels, kernel size, stride, padding, groups]; of pooling layers as [kernel size, stride, padding]; of linear layers as [# input neurons $\rightarrow$ # output neurons]; of flatten operators as [input dimensions $\rightarrow$ output dimensions].

| LAYER | HYPERPARAMETERS |
|---|---|
| STACK | |
| BN1D | |
| CONV1D | $[9 \rightarrow 32, 1, 1, 0, \text{BIAS} = \text{FALSE}]$ |
| CONV1D-RELU-BN1D | $[9 \rightarrow 32, 7, 1, 3]$ |
| G-C-TCN-RELU-BN1D | $[3 \rightarrow 32, 7, 1, 3, 3]$ |
| GP-C-TCN-RELU-BN1D | $[6 \rightarrow 32, 7, 1, 3, 3]$ |
| AVGPOOL1D | $[10, 8, 1]$ |
| BN1D | |
| CONV1D | $[32 \rightarrow 32, 1, 1, 0, \text{BIAS} = \text{FALSE}]$ |
| CONV1D-RELU-BN1D | $[32 \rightarrow 32, 7, 1, 3]$ |
| CONV1D-RELU-BN1D | $[32 \rightarrow 32, 7, 1, 3]$ |
| G-C-TCN-RELU-BN1D | $[4 \rightarrow 32, 7, 1, 3, 8]$ |
| G-C-TCN-RELU-BN1D | $[4 \rightarrow 32, 7, 1, 3, 8]$ |
| GP-C-TCN-RELU-BN1D | $[16 \rightarrow 32, 7, 1, 3, 4]$ |
| GP-C-TCN-RELU-BN1D | $[16 \rightarrow 32, 7, 1, 3, 4]$ |
| AVGPOOL1D | $[6, 4, 1]$ |
| BN1D | |
| CONV1D | $[32 \rightarrow 32, 1, 1, 0, \text{BIAS} = \text{FALSE}]$ |
| CONV1D-RELU-BN1D | $[32 \rightarrow 32, 7, 1, 3]$ |
| CONV1D-RELU-BN1D | $[32 \rightarrow 32, 7, 1, 3]$ |
| G-C-TCN-RELU-BN1D | $[4 \rightarrow 32, 7, 1, 3, 8]$ |
| G-C-TCN-RELU-BN1D | $[4 \rightarrow 32, 7, 1, 3, 8]$ |
| GP-C-TCN-RELU-BN1D | $[16 \rightarrow 32, 7, 1, 3, 4]$ |
| GP-C-TCN-RELU-BN1D | $[16 \rightarrow 32, 7, 1, 3, 4]$ |
| ADAPTIVE AVGPOOL1D | $[25 \rightarrow 16]$ |
| FLATTEN (DEPTH & FEATURE DIM) | $[32 \times 16 \rightarrow 512]$ |
| LINEAR-RELU-BN1D | $[512 \rightarrow 512]$ |
| LINEAR | $[512 \rightarrow 128]$ |

Table 9: **PoNG hyperparameters: Prediction heads.** The parameters of linear layers are denoted as [# input neurons $\rightarrow$ # output neurons].

| LAYER | HYPERPARAMETERS |
|---|---|
| TARGET HEAD $\mathcal{P}^y$ | |
| LINEAR-RELU-LINEAR | $[128 \rightarrow 128]$ |
| LINEAR | $[128 \rightarrow 1]$ |
| AGGREGATE RULE HEAD $\mathcal{P}_1^r$ | |
| SUM | |
| LINEAR-RELU | $[128 \rightarrow 128]$ |
| LINEAR | $[128 \rightarrow d_r]$ |
| TARGET-CONDITIONED RULE HEAD $\mathcal{P}_2^r$ | |
| WEIGHTED SUM | |
| LINEAR | $[128 \rightarrow d_r]$ |

Table 10: **A-I-RAVEN extended regimes.** CS, CT, and ST denote ColorSize, ColorType, and SizeType, resp. P, A, and D3 denote Progression, Arithmetic, and Distribute Three, resp.

| | A/CS | A/CT | A/ST | A/Color-P | A/Color-A | A/Color-D3 |
|---|---|---|---|---|---|---|
| ALANS | 15.1 ($\pm$ 3.3) | 17.7 ($\pm$ 3.2) | 15.7 ($\pm$ 3.2) | 24.8 ($\pm$ 18.8) | 18.3 ($\pm$ 6.6) | 22.4 ($\pm$ 7.7) |
| CPCNet | 33.0 ($\pm$ 5.3) | 25.0 ($\pm$ 0.9) | 24.1 ($\pm$ 1.2) | 50.5 ($\pm$ 0.6) | 45.9 ($\pm$ 2.7) | 37.8 ($\pm$ 0.9) |
| CNN-LSTM | 13.4 ($\pm$ 0.9) | 14.7 ($\pm$ 1.7) | 13.0 ($\pm$ 0.1) | 17.2 ($\pm$ 1.5) | 17.1 ($\pm$ 3.7) | 20.6 ($\pm$ 6.7) |
| CoPINet | 18.3 ($\pm$ 0.3) | 17.2 ($\pm$ 0.1) | 19.7 ($\pm$ 0.7) | 35.8 ($\pm$ 0.6) | 35.2 ($\pm$ 0.5) | 26.9 ($\pm$ 0.5) |
| DRNet | 38.3 ($\pm$ 0.5) | 29.5 ($\pm$ 0.5) | 31.6 ($\pm$ 1.2) | 72.8 ($\pm$ 1.3) | 66.7 ($\pm$ 1.2) | 63.2 ($\pm$ 0.3) |
| MRNet | 18.7 ($\pm$ 1.1) | 20.0 ($\pm$ 2.6) | 28.2 ($\pm$ 0.9) | 34.4 ($\pm$ 3.4) | 35.7 ($\pm$ 5.9) | 18.6 ($\pm$ 0.1) |
| PrAE | 30.0 ($\pm$ 1.1) | 26.7 ($\pm$ 0.7) | 25.6 ($\pm$ 0.8) | 62.3 ($\pm$ 0.9) | 43.0 ($\pm$ 26.5) | 55.1 ($\pm$ 0.8) |
| PredRNet | 31.0 ($\pm$ 1.6) | 28.0 ($\pm$ 0.7) | 27.9 ($\pm$ 0.5) | 62.3 ($\pm$ 2.2) | 56.9 ($\pm$ 1.4) | 48.5 ($\pm$ 0.9) |
| RelBase | 36.6 ($\pm$ 0.8) | 29.7 ($\pm$ 0.6) | 31.1 ($\pm$ 1.0) | 73.0 ($\pm$ 1.8) | 66.2 ($\pm$ 1.0) | 65.7 ($\pm$ 4.6) |
| SCL | 40.8 ($\pm$ 3.2) | 32.0 ($\pm$ 2.3) | **33.5** ($\pm$ 0.7) | 75.6 ($\pm$ 10.1) | 60.0 ($\pm$ 4.1) | 63.9 ($\pm$ 4.3) |
| SRAN | 22.7 ($\pm$ 1.1) | 20.9 ($\pm$ 0.9) | 23.3 ($\pm$ 0.3) | 42.1 ($\pm$ 2.3) | 39.9 ($\pm$ 2.7) | 34.6 ($\pm$ 3.6) |
| STSN | 27.3 ($\pm$ 4.6) | 21.9 ($\pm$ 4.6) | 12.3 ($\pm$ 0.1) | 39.9 ($\pm$ 14.7) | 25.7 ($\pm$ 10.6) | 20.7 ($\pm$ 7.7) |
| WReN | 13.5 ($\pm$ 0.1) | 13.8 ($\pm$ 0.7) | 14.1 ($\pm$ 0.2) | 18.0 ($\pm$ 0.4) | 17.1 ($\pm$ 0.2) | 17.7 ($\pm$ 0.6) |
| PoNG (ours) | **44.7** ($\pm$ 2.1) | **34.3** ($\pm$ 0.8) | 32.1 ($\pm$ 2.1) | **81.4** ($\pm$ 3.1) | **70.0** ($\pm$ 4.1) | **81.3** ($\pm$ 1.6) |

Table 11: I-RAVEN.

| | Test | Val | Test − Val |
|---|---|---|---|
| ALANS | 27.0 ($\pm$ 8.4) | 27.0 ($\pm$ 8.6) | + 0.1 |
| CPCNet | 70.4 ($\pm$ 6.4) | 69.6 ($\pm$ 6.9) | + 0.7 |
| CNN-LSTM | 27.5 ($\pm$ 1.5) | 27.4 ($\pm$ 1.7) | + 0.1 |
| CoPINet | 43.2 ($\pm$ 0.1) | 42.5 ($\pm$ 0.6) | + 0.7 |
| DRNet | 90.9 ($\pm$ 1.1) | 90.8 ($\pm$ 1.2) | + 0.2 |
| MRNet | 86.7 ($\pm$ 2.3) | 86.3 ($\pm$ 2.0) | + 0.5 |
| PrAE | 19.5 ($\pm$ 0.4) | 19.4 ($\pm$ 0.8) | + 0.0 |
| PredRNet | 88.8 ($\pm$ 1.8) | 88.3 ($\pm$ 1.9) | + 0.5 |
| RelBase | 89.6 ($\pm$ 0.6) | 89.5 ($\pm$ 0.5) | + 0.1 |
| SCL | 83.4 ($\pm$ 2.5) | 83.0 ($\pm$ 2.5) | + 0.4 |
| SRAN | 58.2 ($\pm$ 1.6) | 58.0 ($\pm$ 1.3) | + 0.2 |
| STSN | 59.0 ($\pm$ 18.5) | 59.1 ($\pm$ 18.4) | − 0.1 |
| WReN | 18.4 ($\pm$ 0.0) | 18.5 ($\pm$ 0.3) | − 0.1 |
| PoNG (ours) | **95.9** ($\pm$ 0.7) | **95.6** ($\pm$ 0.7) | + 0.3 |

Table 12: I-RAVEN-Mesh.

| | Test | Val | Test − Val |
|---|---|---|---|
| ALANS | 15.9 ($\pm$ 2.6) | 17.1 ($\pm$ 3.6) | − 1.3 |
| CPCNet | 66.6 ($\pm$ 5.1) | 66.5 ($\pm$ 5.4) | + 0.1 |
| CNN-LSTM | 28.9 ($\pm$ 0.4) | 29.3 ($\pm$ 0.6) | − 0.4 |
| CoPINet | 41.1 ($\pm$ 0.3) | 41.3 ($\pm$ 0.2) | − 0.2 |
| DRNet | 83.9 ($\pm$ 2.7) | 84.2 ($\pm$ 2.6) | − 0.3 |
| MRNet | 79.5 ($\pm$ 2.0) | 80.5 ($\pm$ 2.5) | − 1.0 |
| PrAE | 33.2 ($\pm$ 0.4) | 33.0 ($\pm$ 0.9) | + 0.1 |
| PredRNet | 59.2 ($\pm$ 6.4) | 59.3 ($\pm$ 6.9) | − 0.0 |
| RelBase | 84.9 ($\pm$ 4.4) | 85.0 ($\pm$ 4.5) | − 0.1 |
| SCL | 80.9 ($\pm$ 1.5) | 81.0 ($\pm$ 1.5) | − 0.1 |
| SRAN | 57.8 ($\pm$ 0.2) | 58.0 ($\pm$ 0.3) | − 0.2 |
| STSN | 48.7 ($\pm$ 11.5) | 48.8 ($\pm$ 10.9) | − 0.1 |
| WReN | 25.7 ($\pm$ 0.2) | 25.6 ($\pm$ 0.4) | + 0.0 |
| PoNG (ours) | **89.3** ($\pm$ 2.4) | **89.1** ($\pm$ 2.5) | + 0.3 |

Table 13: A/Color.

| | Test | Val | Test − Val |
|---|---|---|---|
| ALANS | 15.2 ($\pm$ 1.4) | 16.4 ($\pm$ 2.1) | − 1.2 |
| CPCNet | 51.2 ($\pm$ 3.8) | 77.0 ($\pm$ 6.1) | −25.7 |
| CNN-LSTM | 17.0 ($\pm$ 3.1) | 31.0 ($\pm$ 4.4) | −13.9 |
| CoPINet | 32.5 ($\pm$ 0.2) | 49.9 ($\pm$ 0.7) | −17.4 |
| DRNet | 70.0 ($\pm$ 1.6) | 95.4 ($\pm$ 0.2) | −25.4 |
| MRNet | 33.6 ($\pm$ 8.2) | 86.2 ($\pm$ 6.6) | −52.6 |
| PrAE | 47.9 ($\pm$ 0.9) | 60.9 ($\pm$ 1.4) | −13.0 |
| PredRNet | 59.4 ($\pm$ 1.0) | 92.2 ($\pm$ 1.0) | −32.9 |
| RelBase | 67.4 ($\pm$ 2.7) | 95.2 ($\pm$ 0.4) | −27.8 |
| SCL | 65.1 ($\pm$ 2.0) | 84.4 ($\pm$ 0.5) | −19.2 |
| SRAN | 38.3 ($\pm$ 1.0) | 63.7 ($\pm$ 0.3) | −25.4 |
| STSN | 39.3 ($\pm$ 6.9) | 71.3 ($\pm$ 17.0) | −32.0 |
| WReN | 16.9 ($\pm$ 0.5) | 23.2 ($\pm$ 0.8) | − 6.3 |
| PoNG (ours) | **80.3** ($\pm$ 4.3) | **96.9** ($\pm$ 0.4) | −16.6 |

Table 14: A/Position.

| | Test | Val | Test − Val |
|---|---|---|---|
| ALANS | 16.0 ($\pm$ 1.0) | 15.2 ($\pm$ 1.3) | + 0.8 |
| CPCNet | 68.3 ($\pm$ 4.0) | 90.6 ($\pm$ 5.3) | −22.3 |
| CNN-LSTM | 24.0 ($\pm$ 2.9) | 36.4 ($\pm$ 3.7) | −12.4 |
| CoPINet | 41.3 ($\pm$ 1.6) | 54.7 ($\pm$ 1.7) | −13.4 |
| DRNet | 77.5 ($\pm$ 0.9) | 97.8 ($\pm$ 0.1) | −20.2 |
| MRNet | 62.6 ($\pm$ 2.6) | 94.4 ($\pm$ 7.0) | −31.8 |
| PrAE | 68.2 ($\pm$ 3.3) | 80.1 ($\pm$ 3.4) | −12.0 |
| PredRNet | 73.7 ($\pm$ 0.7) | 97.4 ($\pm$ 0.5) | −23.6 |
| RelBase | 76.6 ($\pm$ 0.3) | 97.0 ($\pm$ 0.2) | −20.4 |
| SCL | 76.7 ($\pm$ 7.1) | 94.7 ($\pm$ 5.3) | −18.0 |
| SRAN | 56.9 ($\pm$ 0.7) | 75.6 ($\pm$ 1.4) | −18.8 |
| STSN | 36.1 ($\pm$ 19.9) | 50.7 ($\pm$ 27.3) | −14.6 |
| WReN | 17.3 ($\pm$ 0.4) | 23.3 ($\pm$ 0.5) | − 6.0 |
| PoNG (ours) | **79.3** ($\pm$ 0.7) | **98.2** ($\pm$ 0.1) | −18.9 |

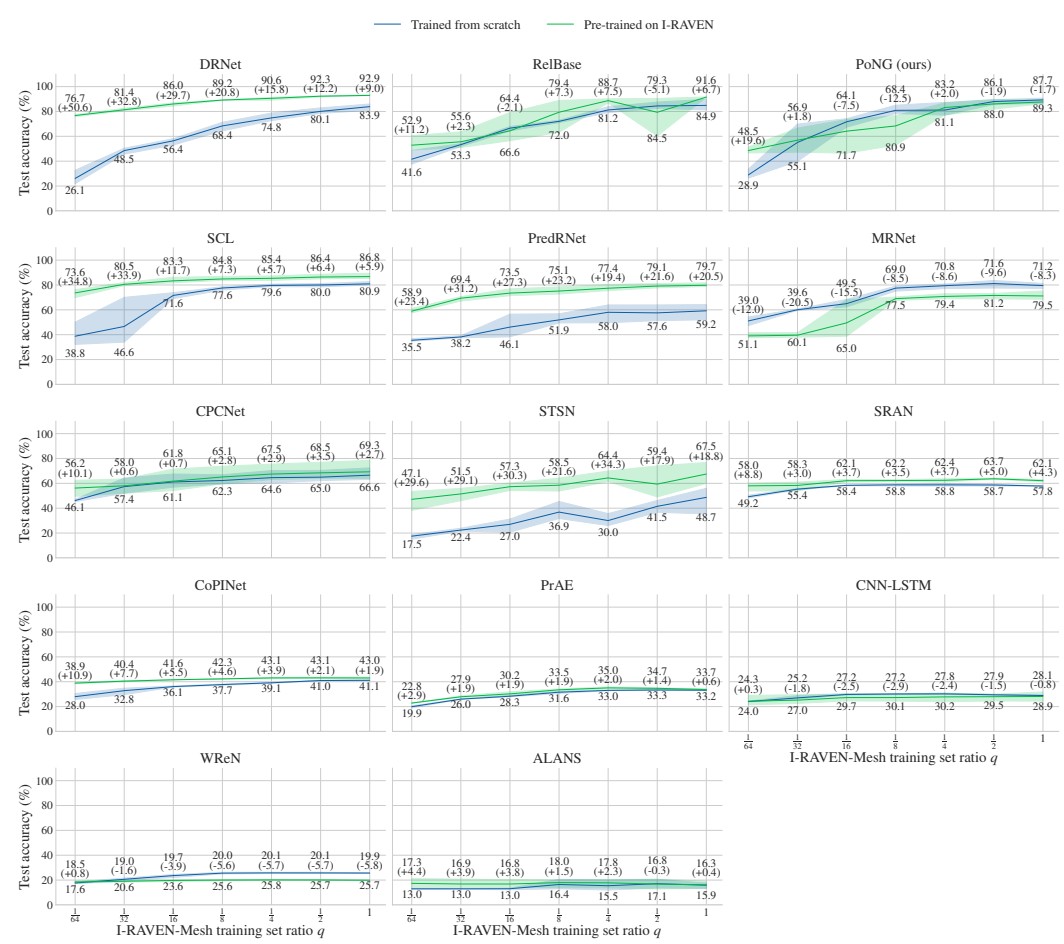

Figure 12: **Transfer learning.** Mean and standard deviation of test accuracy on I-RAVEN-Mesh across three random seeds. Models were trained in two setups: 1) from scratch on I-RAVEN-Mesh with variable sample size; 2) pre-trained on full I-RAVEN and fine-tuned on I-RAVEN-Mesh with variable sample size. Results for setups 1) and 2) are shown below and above the plot lines, resp. This figure extends Fig. 6 with all models considered in the paper.

Table 15: A/Size.

|  | Test | Val | Test − Val |
|---|---|---|---|
| ALANS | 23.3 (± 6.5) | 24.6 (± 10.6) | − 1.2 |
| CPCNet | 43.5 (± 3.5) | 79.2 (± 2.1) | −35.6 |
| CNN-LSTM | 13.6 (± 1.4) | 37.2 (± 3.1) | −23.6 |
| CoPINet | 21.8 (± 0.2) | 60.7 (± 0.1) | −38.9 |
| DRNet | 54.3 (± 3.0) | 93.1 (± 0.5) | −38.8 |
| MRNet | 20.6 (± 5.0) | 88.1 (± 2.5) | −67.5 |
| PrAE | 41.3 (± 1.8) | 59.7 (± 1.8) | −18.4 |
| PredRNet | 47.5 (± 1.3) | 92.5 (± 0.8) | −45.0 |
| RelBase | 51.1 (± 2.4) | 92.9 (± 0.4) | −41.8 |
| SCL | 65.6 (± 2.4) | 94.0 (± 2.6) | −28.4 |
| SRAN | 34.4 (± 3.0) | 78.1 (± 0.2) | −43.7 |
| STSN | 38.4 (± 16.6) | 68.9 (± 34.1) | −30.5 |
| WReN | 12.4 (± 0.5) | 29.5 (± 0.5) | −17.1 |
| PoNG (ours) | **73.5** (± 3.1) | **97.1** (± 0.5) | −23.6 |

Table 16: A/Type.

|  | Test | Val | Test − Val |
|---|---|---|---|
| ALANS | 19.0 (± 3.4) | 25.2 (± 5.2) | − 6.3 |
| CPCNet | 38.6 (± 4.3) | 85.2 (± 3.3) | −46.7 |
| CNN-LSTM | 14.5 (± 0.8) | 38.5 (± 0.8) | −24.0 |
| CoPINet | 19.8 (± 0.9) | 58.8 (± 2.6) | −39.0 |
| DRNet | 44.3 (± 0.8) | 93.9 (± 0.3) | −49.6 |
| MRNet | 19.4 (± 0.3) | 93.0 (± 0.7) | −73.7 |
| PrAE | 37.0 (± 1.7) | 61.1 (± 1.4) | −24.1 |
| PredRNet | 40.2 (± 1.3) | 93.9 (± 0.1) | −53.7 |
| RelBase | 44.1 (± 1.0) | 93.4 (± 0.2) | −49.3 |
| SCL | 49.5 (± 1.8) | 95.9 (± 0.7) | −46.4 |
| SRAN | 30.7 (± 2.2) | 78.8 (± 0.7) | −48.1 |
| STSN | 39.1 (± 5.0) | 66.2 (± 17.7) | −27.0 |
| WReN | 15.1 (± 0.7) | 23.4 (± 0.6) | − 8.2 |
| PoNG (ours) | **59.4** (± 6.9) | **96.1** (± 0.8) | −36.7 |

Table 17: A/ColorSize.

|  | Test | Val | Test − Val |
|---|---|---|---|
| ALANS | 15.1 (± 3.3) | 16.4 (± 4.3) | − 1.3 |
| CPCNet | 33.0 (± 5.3) | 86.1 (± 1.4) | −53.1 |
| CNN-LSTM | 13.4 (± 0.9) | 53.1 (± 6.1) | −39.8 |
| CoPINet | 18.3 (± 0.3) | 71.7 (± 0.3) | −53.4 |
| DRNet | 38.3 (± 0.5) | 95.9 (± 0.6) | −57.6 |
| MRNet | 18.7 (± 1.1) | 92.8 (± 2.1) | −74.0 |
| PrAE | 30.0 (± 1.1) | 63.2 (± 2.3) | −33.1 |
| PredRNet | 31.0 (± 1.6) | 95.4 (± 0.4) | −64.4 |
| RelBase | 36.6 (± 0.8) | 95.2 (± 0.6) | −58.7 |
| SCL | 40.8 (± 3.2) | 94.2 (± 1.7) | −53.4 |
| SRAN | 22.7 (± 1.1) | 84.3 (± 0.7) | −61.7 |
| STSN | 27.3 (± 4.6) | 84.5 (± 12.7) | −57.2 |
| WReN | 13.5 (± 0.1) | 43.4 (± 1.4) | −29.9 |
| PoNG (ours) | **44.7** (± 2.1) | **97.3** (± 0.5) | −52.6 |

Table 18: A/ColorType.

|  | Test | Val | Test − Val |
|---|---|---|---|
| ALANS | 17.7 (± 3.2) | 20.4 (± 6.2) | − 2.7 |
| CPCNet | 25.0 (± 0.9) | 84.4 (± 2.7) | −59.4 |
| CNN-LSTM | 14.7 (± 1.7) | 53.3 (± 5.9) | −38.7 |
| CoPINet | 17.2 (± 0.1) | 72.8 (± 0.5) | −55.6 |
| DRNet | 29.5 (± 0.5) | 96.0 (± 0.4) | −66.4 |
| MRNet | 20.0 (± 2.6) | 91.2 (± 4.2) | −71.2 |
| PrAE | 26.7 (± 0.7) | 58.8 (± 2.6) | −32.1 |
| PredRNet | 28.0 (± 0.7) | 91.6 (± 0.6) | −63.6 |
| RelBase | 29.7 (± 0.6) | 93.0 (± 4.0) | −63.3 |
| SCL | 32.0 (± 2.3) | 96.4 (± 0.4) | −64.4 |
| SRAN | 20.9 (± 0.9) | 84.4 (± 1.9) | −63.5 |
| STSN | 21.9 (± 4.6) | 76.2 (± 22.2) | −54.3 |
| WReN | 13.8 (± 0.7) | 41.3 (± 1.0) | −27.5 |
| PoNG (ours) | **34.3** (± 0.8) | **96.6** (± 0.5) | −62.3 |

Table 19: A/SizeType.

|  | Test | Val | Test − Val |
|---|---|---|---|
| ALANS | 15.7 (± 3.2) | 21.6 (± 11.7) | − 5.9 |
| CPCNet | 24.1 (± 1.2) | 87.0 (± 1.8) | −62.9 |
| CNN-LSTM | 13.0 (± 0.1) | 53.6 (± 0.2) | −40.6 |
| CoPINet | 19.7 (± 0.7) | 72.8 (± 0.3) | −53.1 |
| DRNet | 31.6 (± 1.2) | 95.6 (± 0.2) | −64.0 |
| MRNet | 28.2 (± 0.9) | **97.7** (± 0.2) | −69.5 |
| PrAE | 25.6 (± 0.8) | 65.0 (± 3.7) | −39.4 |
| PredRNet | 27.9 (± 0.5) | 94.8 (± 0.3) | −66.9 |
| RelBase | 31.1 (± 1.0) | 94.6 (± 1.4) | −63.5 |
| SCL | **33.5** (± 0.7) | 97.2 (± 0.6) | −63.7 |
| SRAN | 23.3 (± 0.3) | 88.4 (± 0.3) | −65.1 |
| STSN | 12.3 (± 0.1) | 12.5 (± 0.5) | − 0.2 |
| WReN | 14.1 (± 0.2) | 50.9 (± 0.7) | −36.8 |
| PoNG (ours) | 32.1 (± 2.1) | 96.6 (± 0.9) | −64.5 |

Table 20: A/Color-Progression.

|  | Test | Val | Test − Val |
|---|---|---|---|
| ALANS | 24.8 (± 18.8) | 26.0 (± 20.2) | − 1.2 |
| CPCNet | 50.5 (± 0.6) | 75.5 (± 0.7) | −25.0 |
| CNN-LSTM | 17.2 (± 1.5) | 29.6 (± 2.7) | −12.4 |
| CoPINet | 35.8 (± 0.6) | 48.9 (± 0.5) | −13.1 |
| DRNet | 72.8 (± 1.3) | 96.1 (± 0.5) | −23.3 |
| MRNet | 34.4 (± 3.4) | 93.0 (± 2.7) | −58.6 |
| PrAE | 62.3 (± 0.9) | 65.1 (± 3.7) | − 2.8 |
| PredRNet | 62.3 (± 2.2) | 91.0 (± 2.5) | −28.7 |
| RelBase | 73.0 (± 1.8) | 95.0 (± 0.8) | −22.0 |
| SCL | 75.6 (± 10.1) | 88.8 (± 7.1) | −13.2 |
| SRAN | 42.1 (± 2.3) | 64.4 (± 1.8) | −22.4 |
| STSN | 39.9 (± 14.7) | 70.7 (± 28.9) | −30.8 |
| WReN | 18.0 (± 0.4) | 18.8 (± 0.4) | − 0.7 |
| PoNG (ours) | **81.4** (± 3.1) | **97.4** (± 0.3) | −16.0 |

Table 21: A/Color-Arithmetic.

|  | Test | Val | Test − Val |
|---|---|---|---|
| ALANS | 18.3 (± 6.6) | 19.2 (± 6.5) | − 0.9 |
| CPCNet | 45.9 (± 2.7) | 74.3 (± 1.8) | −28.4 |
| CNN-LSTM | 17.1 (± 3.7) | 26.0 (± 4.5) | − 8.9 |
| CoPINet | 35.2 (± 0.5) | 45.5 (± 0.8) | −10.4 |
| DRNet | 66.7 (± 1.2) | 93.5 (± 0.5) | −26.8 |
| MRNet | 35.7 (± 5.9) | 88.8 (± 5.6) | −53.1 |
| PrAE | 43.0 (± 26.5) | 43.4 (± 26.7) | − 0.4 |
| PredRNet | 56.9 (± 1.4) | 89.5 (± 0.4) | −32.6 |
| RelBase | 66.2 (± 1.0) | 93.5 (± 0.2) | −27.3 |
| SCL | 60.0 (± 4.1) | 85.0 (± 1.5) | −25.1 |
| SRAN | 39.9 (± 2.7) | 61.7 (± 3.2) | −21.8 |
| STSN | 25.7 (± 10.6) | 39.5 (± 25.8) | −13.7 |
| WReN | 17.1 (± 0.2) | 18.2 (± 0.3) | − 1.1 |
| PoNG (ours) | **70.0** (± 4.1) | **96.3** (± 0.3) | −26.3 |

Table 22: A/Color-DistributeThree.

|  | Test | Val | Test − Val |
|---|---|---|---|
| ALANS | 22.4 (± 7.7) | 21.5 (± 7.0) | + 0.9 |
| CPCNet | 37.8 (± 0.9) | 74.5 (± 0.5) | −36.7 |
| CNN-LSTM | 20.6 (± 6.7) | 24.9 (± 0.5) | − 4.3 |
| CoPINet | 26.9 (± 0.5) | 48.7 (± 0.5) | −21.8 |
| DRNet | 63.2 (± 0.3) | 95.1 (± 0.2) | −31.9 |
| MRNet | 18.6 (± 0.1) | 92.6 (± 0.4) | −74.0 |
| PrAE | 55.1 (± 0.8) | 63.4 (± 3.7) | − 8.3 |
| PredRNet | 48.5 (± 0.9) | 89.7 (± 0.9) | −41.2 |
| RelBase | 65.7 (± 4.6) | 93.7 (± 1.6) | −28.0 |
| SCL | 63.9 (± 4.3) | 83.4 (± 1.4) | −19.5 |
| SRAN | 34.6 (± 3.6) | 64.7 (± 2.3) | −30.1 |
| STSN | 20.7 (± 7.7) | 53.5 (± 23.2) | −32.8 |
| WReN | 17.7 (± 0.6) | 18.3 (± 0.6) | − 0.6 |
| PoNG (ours) | **81.3** (± 1.6) | **96.8** (± 0.9) | −15.5 |

Table 23: I-RAVEN. Results from Table 2 extended to each matrix configuration.

| | Mean | Center | 2x2Grid | 3x3Grid | L-R | U-D | O-IC | O-IG |
|---|---|---|---|---|---|---|---|---|
| ALANS | 27.0 (±8.4) | 28.8 (±9.2) | 25.6 (±5.2) | 26.7 (±7.7) | 32.1 (±13.2) | 31.9 (±12.0) | 24.3 (±8.5) | 19.8 (±3.8) |
| CPCNet | 70.4 (±6.4) | 85.0 (±10.9) | 53.1 (±10.9) | 45.2 (±7.0) | 89.3 (±4.4) | 89.9 (±5.1) | 84.3 (±1.3) | 46.0 (±5.7) |
| CNN-LSTM | 27.5 (±1.5) | 41.2 (±4.3) | 27.1 (±3.1) | 24.8 (±3.3) | 23.4 (±1.9) | 22.5 (±0.2) | 28.3 (±2.0) | 25.6 (±0.7) |
| CoPINet | 43.2 (±0.1) | 51.8 (±0.7) | 34.0 (±0.6) | 29.9 (±0.6) | 47.2 (±0.6) | 49.9 (±0.6) | 49.2 (±0.9) | 40.3 (±1.2) |
| DRNet | 90.9 (±1.1) | 99.3 (±0.2) | 91.2 (±1.5) | 83.6 (±1.1) | 95.8 (±0.8) | 98.2 (±0.4) | 98.1 (±0.7) | 70.3 (±3.8) |
| MRNet | 86.7 (±2.3) | 99.8 (±0.1) | 82.2 (±10.0) | 72.3 (±11.1) | 97.8 (±2.0) | 97.9 (±1.3) | 94.4 (±3.0) | 62.9 (±0.8) |
| PrAE | 19.5 (±0.4) | 20.2 (±0.7) | 36.8 (±1.0) | 17.3 (±1.8) | 16.4 (±0.7) | 15.3 (±0.7) | 15.8 (±1.4) | 14.6 (±0.6) |
| PredRNet | 88.8 (±1.8) | 98.7 (±0.2) | 93.4 (±1.3) | 80.3 (±4.8) | 98.5 (±0.5) | 97.7 (±0.3) | 97.7 (±1.0) | 55.3 (±7.3) |
| RelBase | 89.6 (±0.6) | 99.1 (±0.2) | 90.6 (±0.7) | 82.2 (±1.4) | 95.2 (±0.3) | 97.7 (±0.5) | 98.0 (±0.5) | 64.2 (±2.1) |
| SCL | 83.4 (±2.5) | 98.6 (±0.2) | 81.6 (±3.3) | 73.1 (±0.7) | 86.7 (±5.3) | 86.3 (±5.7) | 88.4 (±3.2) | 69.4 (±0.5) |
| SRAN | 58.2 (±1.6) | 80.7 (±2.8) | 46.7 (±0.8) | 39.7 (±1.2) | 68.1 (±2.9) | 66.5 (±3.1) | 62.6 (±0.3) | 42.9 (±1.0) |
| STSN | 59.0 (±18.5) | 76.1 (±22.7) | 53.7 (±15.9) | 48.5 (±13.7) | 65.1 (±24.5) | 64.1 (±24.8) | 65.5 (±22.3) | 41.7 (±8.5) |
| WReN | 18.4 (±0.0) | 24.5 (±3.2) | 17.2 (±1.1) | 17.3 (±0.9) | 15.1 (±1.0) | 16.8 (±1.0) | 18.7 (±0.5) | 19.6 (±0.4) |
| PoNG (ours) | 95.9 (±0.7) | 99.5 (±0.1) | 97.8 (±0.9) | 91.2 (±1.2) | 98.7 (±0.5) | 98.7 (±0.6) | 98.8 (±0.3) | 86.5 (±2.1) |

Table 24: I-RAVEN-Mesh. Results from Table 2 extended to each matrix configuration.

| | Mean | Center | 2x2Grid | 3x3Grid | L-R | U-D | O-IC | O-IG |
|---|---|---|---|---|---|---|---|---|
| ALANS | 15.9 (±2.6) | 16.3 (±2.8) | 15.8 (±2.9) | 16.1 (±2.7) | 16.0 (±3.5) | 17.1 (±3.6) | 15.6 (±1.9) | 14.2 (±0.6) |
| CPCNet | 66.6 (±5.1) | 73.2 (±11.6) | 53.4 (±5.0) | 50.1 (±2.9) | 78.4 (±3.9) | 76.9 (±4.4) | 74.9 (±3.2) | 58.9 (±5.4) |
| CNN-LSTM | 28.9 (±0.4) | 30.5 (±0.9) | 27.4 (±0.7) | 28.4 (±0.5) | 28.9 (±0.5) | 29.0 (±0.8) | 28.8 (±0.3) | 29.1 (±1.5) |
| CoPINet | 41.1 (±0.3) | 41.5 (±0.3) | 38.2 (±0.3) | 34.6 (±0.2) | 42.2 (±1.7) | 42.4 (±0.2) | 46.0 (±0.3) | 42.9 (±0.5) |
| DRNet | 83.9 (±2.7) | 93.7 (±1.0) | 77.5 (±6.6) | 71.4 (±4.2) | 88.2 (±0.8) | 90.9 (±1.3) | 89.6 (±0.9) | 76.0 (±5.3) |
| MRNet | 79.5 (±2.0) | 93.2 (±4.9) | 65.4 (±6.2) | 59.4 (±5.1) | 90.1 (±4.2) | 91.2 (±3.8) | 89.5 (±1.6) | 67.6 (±1.7) |
| PrAE | 33.2 (±0.4) | 38.1 (±1.0) | 39.1 (±0.1) | 19.9 (±0.9) | 41.3 (±1.7) | 41.7 (±0.7) | 28.4 (±1.5) | 23.7 (±1.0) |
| PredRNet | 59.2 (±6.4) | 67.9 (±5.4) | 50.7 (±4.3) | 47.2 (±2.9) | 65.2 (±8.9) | 63.0 (±9.3) | 65.1 (±9.5) | 55.0 (±5.0) |
| RelBase | 84.9 (±4.4) | 92.5 (±2.3) | 81.5 (±5.8) | 75.5 (±5.8) | 88.4 (±3.4) | 90.4 (±2.8) | 90.3 (±5.0) | 75.3 (±6.7) |
| SCL | 80.9 (±1.5) | 88.5 (±2.5) | 77.6 (±1.2) | 73.5 (±1.8) | 83.1 (±0.0) | 82.8 (±0.6) | 86.7 (±0.8) | 74.1 (±5.5) |
| SRAN | 57.8 (±0.2) | 65.7 (±0.3) | 46.6 (±0.9) | 45.1 (±0.9) | 64.0 (±0.2) | 66.1 (±1.3) | 63.8 (±0.4) | 53.5 (±0.6) |
| STSN | 48.7 (±11.5) | 63.8 (±20.2) | 45.2 (±9.7) | 43.4 (±7.3) | 43.9 (±6.9) | 44.0 (±7.9) | 52.8 (±17.2) | 49.0 (±12.6) |
| WReN | 25.7 (±0.2) | 26.4 (±0.5) | 24.5 (±0.4) | 25.7 (±0.6) | 25.7 (±0.3) | 24.9 (±1.4) | 25.9 (±0.1) | 26.3 (±0.2) |
| PoNG (ours) | 89.3 (±2.4) | 91.6 (±3.5) | 89.6 (±2.3) | 82.8 (±2.7) | 90.8 (±2.4) | 91.7 (±2.3) | 91.0 (±1.8) | 87.7 (±2.2) |

Table 25: A/Color. Results from Table 2 extended to each matrix configuration.

| | Mean | Center | 2x2Grid | 3x3Grid | L-R | U-D | O-IC | O-IG |
|---|---|---|---|---|---|---|---|---|
| ALANS | 15.2 (±1.4) | 14.7 (±2.3) | 16.7 (±2.0) | 15.1 (±1.5) | 15.6 (±2.8) | 15.7 (±2.0) | 14.6 (±1.8) | 14.0 (±1.5) |
| CPCNet | 51.2 (±3.8) | 48.3 (±6.8) | 41.1 (±6.7) | 38.9 (±2.6) | 57.4 (±2.7) | 57.5 (±2.0) | 67.1 (±2.4) | 48.0 (±5.7) |
| CNN-LSTM | 17.0 (±3.1) | 17.9 (±4.6) | 17.3 (±2.7) | 16.3 (±2.3) | 15.8 (±1.8) | 15.9 (±2.6) | 18.3 (±4.6) | 17.9 (±3.6) |
| CoPINet | 32.5 (±0.2) | 33.0 (±0.8) | 28.9 (±0.7) | 27.0 (±0.2) | 30.0 (±0.4) | 30.7 (±0.7) | 39.2 (±0.6) | 38.7 (±1.7) |
| DRNet | 70.0 (±1.6) | 66.9 (±4.0) | 65.9 (±1.2) | 63.9 (±1.1) | 68.1 (±1.7) | 70.1 (±3.2) | 82.7 (±2.0) | 72.5 (±1.6) |
| MRNet | 33.6 (±8.2) | 27.1 (±7.1) | 39.6 (±1.4) | 39.4 (±3.0) | 21.1 (±16.4) | 20.3 (±17.3) | 39.8 (±16.9) | 48.4 (±5.0) |
| PrAE | 47.9 (±0.9) | 50.0 (±1.5) | 57.7 (±1.8) | 36.7 (±2.2) | 61.8 (±1.6) | 60.8 (±1.1) | 37.8 (±0.5) | 30.3 (±1.5) |
| PredRNet | 59.4 (±1.0) | 52.9 (±1.0) | 61.3 (±0.3) | 56.9 (±2.2) | 58.0 (±0.7) | 57.8 (±1.3) | 69.9 (±0.3) | 59.5 (±6.0) |
| RelBase | 67.4 (±2.7) | 62.8 (±4.5) | 66.8 (±2.6) | 63.4 (±2.4) | 66.4 (±3.4) | 66.2 (±3.8) | 76.2 (±3.2) | 70.2 (±2.4) |
| SCL | 65.1 (±2.0) | 71.3 (±5.2) | 66.2 (±2.1) | 57.6 (±1.9) | 57.5 (±5.0) | 56.6 (±4.9) | 74.0 (±0.7) | 73.5 (±2.6) |
| SRAN | 38.3 (±1.0) | 40.1 (±2.4) | 35.0 (±0.7) | 31.5 (±1.4) | 35.9 (±3.5) | 36.2 (±2.6) | 47.4 (±0.7) | 42.0 (±0.6) |
| STSN | 39.3 (±6.9) | 39.5 (±1.8) | 39.9 (±7.6) | 38.6 (±7.0) | 31.9 (±9.1) | 30.0 (±7.4) | 49.6 (±8.0) | 45.3 (±9.0) |
| WReN | 16.9 (±0.5) | 18.7 (±0.6) | 17.1 (±0.4) | 16.2 (±0.7) | 15.3 (±0.5) | 15.7 (±1.3) | 17.7 (±0.3) | 17.5 (±0.8) |
| PoNG (ours) | 80.3 (±4.3) | 84.4 (±10.1) | 85.4 (±6.8) | 80.3 (±5.1) | 72.3 (±3.7) | 71.3 (±3.8) | 88.9 (±3.1) | 79.0 (±3.7) |

Table 26: `A/Position`. Results from Table 2 extended to each matrix configuration.

| | Mean | Center | 2x2Grid | 3x3Grid | L-R | U-D | O-IC | O-IG |
|---|---|---|---|---|---|---|---|---|
| ALANS | 16.0 ($\pm$1.0) | 15.0 ($\pm$1.9) | 18.1 ($\pm$0.8) | 16.8 ($\pm$1.3) | 16.7 ($\pm$1.6) | 16.5 ($\pm$1.0) | 15.2 ($\pm$0.5) | 14.0 ($\pm$2.6) |
| CPCNet | 68.3 ($\pm$4.0) | 89.4 ($\pm$8.8) | 29.1 ($\pm$2.3) | 30.1 ($\pm$3.6) | 96.1 ($\pm$3.1) | 96.5 ($\pm$2.8) | 94.3 ($\pm$2.6) | 42.6 ($\pm$6.2) |
| CNN-LSTM | 24.0 ($\pm$2.9) | 43.8 ($\pm$5.6) | 13.8 ($\pm$1.8) | 14.5 ($\pm$1.2) | 25.4 ($\pm$1.9) | 26.0 ($\pm$2.9) | 29.1 ($\pm$5.1) | 15.1 ($\pm$3.2) |
| CoPINet | 41.3 ($\pm$1.6) | 51.2 ($\pm$2.6) | 22.2 ($\pm$0.7) | 22.2 ($\pm$0.7) | 53.1 ($\pm$2.4) | 52.8 ($\pm$2.7) | 53.3 ($\pm$1.6) | 34.5 ($\pm$1.6) |
| DRNet | 77.5 ($\pm$0.9) | 99.4 ($\pm$0.1) | 41.8 ($\pm$3.3) | 41.3 ($\pm$1.6) | 97.3 ($\pm$0.6) | 98.6 ($\pm$0.3) | **99.2** ($\pm$0.1) | **65.4** ($\pm$1.2) |
| MRNet | 62.6 ($\pm$2.6) | 92.0 ($\pm$13.7) | 19.5 ($\pm$3.2) | 18.2 ($\pm$4.5) | 98.1 ($\pm$2.5) | 97.1 ($\pm$3.7) | 96.4 ($\pm$4.1) | 16.9 ($\pm$0.8) |
| PrAE | 68.2 ($\pm$3.3) | 86.0 ($\pm$5.5) | **56.5** ($\pm$1.9) | **50.2** ($\pm$1.9) | 92.0 ($\pm$4.7) | 92.3 ($\pm$4.5) | 62.0 ($\pm$5.1) | 38.1 ($\pm$1.6) |
| PredRNet | 73.7 ($\pm$0.7) | 99.2 ($\pm$0.2) | 36.2 ($\pm$1.4) | 36.2 ($\pm$1.5) | 98.7 ($\pm$0.7) | **98.9** ($\pm$0.1) | 98.6 ($\pm$0.2) | 48.2 ($\pm$1.1) |
| RelBase | 76.6 ($\pm$0.3) | 99.1 ($\pm$0.1) | 39.7 ($\pm$1.1) | 39.9 ($\pm$2.5) | 96.1 ($\pm$0.3) | 98.0 ($\pm$0.3) | 98.8 ($\pm$0.0) | 64.3 ($\pm$0.8) |
| SCL | 76.7 ($\pm$7.1) | 99.2 ($\pm$0.8) | 51.6 ($\pm$7.6) | 43.4 ($\pm$9.7) | 93.4 ($\pm$8.9) | 93.9 ($\pm$8.1) | 95.6 ($\pm$5.3) | 60.0 ($\pm$9.5) |
| SRAN | 56.9 ($\pm$0.7) | 80.0 ($\pm$5.1) | 31.2 ($\pm$0.6) | 30.3 ($\pm$0.7) | 73.9 ($\pm$1.2) | 74.0 ($\pm$2.5) | 69.9 ($\pm$0.6) | 39.0 ($\pm$4.4) |
| STSN | 36.1 ($\pm$19.9) | 54.1 ($\pm$22.8) | 19.5 ($\pm$6.6) | 20.5 ($\pm$9.0) | 39.7 ($\pm$32.8) | 41.7 ($\pm$30.0) | 48.3 ($\pm$30.2) | 30.1 ($\pm$10.6) |
| WReN | 17.3 ($\pm$0.4) | 21.1 ($\pm$1.5) | 15.6 ($\pm$0.3) | 15.9 ($\pm$1.2) | 15.4 ($\pm$1.1) | 16.3 ($\pm$0.5) | 19.7 ($\pm$1.2) | 16.8 ($\pm$1.1) |
| PoNG (ours) | **79.3** ($\pm$0.7) | **99.6** ($\pm$0.1) | 54.5 ($\pm$1.0) | 44.0 ($\pm$2.8) | **98.8** ($\pm$0.1) | 98.9 ($\pm$0.2) | 98.7 ($\pm$0.2) | 60.6 ($\pm$2.8) |

Table 27: `A/Size`. Results from Table 2 extended to each matrix configuration.

| | Mean | Center | 2x2Grid | 3x3Grid | L-R | U-D | O-IC | O-IG |
|---|---|---|---|---|---|---|---|---|
| ALANS | 23.3 ($\pm$6.5) | 23.9 ($\pm$7.4) | 23.6 ($\pm$4.9) | 22.1 ($\pm$5.8) | 26.6 ($\pm$8.1) | 26.9 ($\pm$8.6) | 22.0 ($\pm$7.1) | 18.4 ($\pm$3.9) |
| CPCNet | 43.5 ($\pm$3.5) | 48.8 ($\pm$6.3) | 38.2 ($\pm$3.9) | 36.3 ($\pm$1.8) | 51.8 ($\pm$4.1) | 50.6 ($\pm$1.7) | 47.8 ($\pm$6.2) | 31.0 ($\pm$2.8) |
| CNN-LSTM | 13.6 ($\pm$1.4) | 15.8 ($\pm$2.4) | 16.6 ($\pm$1.3) | 15.0 ($\pm$2.6) | 12.7 ($\pm$1.3) | 12.3 ($\pm$1.1) | 11.2 ($\pm$1.9) | 11.5 ($\pm$1.1) |
| CoPINet | 21.8 ($\pm$0.2) | 22.9 ($\pm$1.0) | 25.9 ($\pm$1.2) | 27.0 ($\pm$0.3) | 20.1 ($\pm$0.9) | 22.1 ($\pm$0.5) | 15.0 ($\pm$0.5) | 19.4 ($\pm$0.6) |
| DRNet | 54.3 ($\pm$3.0) | 63.0 ($\pm$4.8) | 56.5 ($\pm$1.3) | 52.7 ($\pm$1.5) | 52.4 ($\pm$2.8) | 56.6 ($\pm$3.9) | 48.9 ($\pm$7.5) | 50.2 ($\pm$5.5) |
| MRNet | 20.6 ($\pm$5.0) | 20.8 ($\pm$12.1) | 37.6 ($\pm$6.3) | 35.1 ($\pm$5.7) | 8.1 ($\pm$4.1) | 8.7 ($\pm$2.7) | 12.1 ($\pm$5.1) | 21.7 ($\pm$3.1) |
| PrAE | 41.3 ($\pm$1.8) | 38.9 ($\pm$4.1) | 49.5 ($\pm$1.6) | 34.4 ($\pm$3.8) | 54.2 ($\pm$1.7) | 53.9 ($\pm$3.5) | 31.4 ($\pm$2.2) | 27.0 ($\pm$1.4) |
| PredRNet | 47.5 ($\pm$1.3) | 55.4 ($\pm$3.9) | 52.9 ($\pm$0.5) | 50.3 ($\pm$0.3) | 48.2 ($\pm$1.3) | 48.1 ($\pm$1.0) | 39.0 ($\pm$0.3) | 38.1 ($\pm$5.2) |
| RelBase | 51.1 ($\pm$2.4) | 59.3 ($\pm$1.8) | 54.3 ($\pm$0.7) | 52.8 ($\pm$0.5) | 50.3 ($\pm$3.0) | 52.8 ($\pm$1.5) | 42.0 ($\pm$7.1) | 46.5 ($\pm$4.7) |
| SCL | 65.6 ($\pm$2.4) | 66.2 ($\pm$3.9) | 72.1 ($\pm$5.5) | 67.0 ($\pm$4.5) | 61.7 ($\pm$0.5) | 62.0 ($\pm$0.3) | 68.6 ($\pm$3.7) | 61.5 ($\pm$1.9) |
| SRAN | 34.4 ($\pm$3.0) | 39.4 ($\pm$7.7) | 35.9 ($\pm$2.0) | 35.9 ($\pm$1.3) | 35.8 ($\pm$2.9) | 37.3 ($\pm$4.6) | 30.4 ($\pm$6.1) | 26.2 ($\pm$2.7) |
| STSN | 38.4 ($\pm$16.6) | 40.6 ($\pm$14.9) | 42.0 ($\pm$21.7) | 38.4 ($\pm$18.5) | 36.8 ($\pm$16.8) | 38.3 ($\pm$16.1) | 40.1 ($\pm$16.7) | 33.1 ($\pm$13.3) |
| WReN | 12.4 ($\pm$0.5) | 13.8 ($\pm$0.5) | 14.1 ($\pm$0.1) | 15.0 ($\pm$0.6) | 13.3 ($\pm$0.9) | 13.3 ($\pm$0.9) | 8.2 ($\pm$1.0) | 9.2 ($\pm$0.9) |
| PoNG (ours) | **73.5** ($\pm$3.1) | **84.0** ($\pm$2.9) | **81.1** ($\pm$8.1) | **75.2** ($\pm$8.6) | **67.7** ($\pm$3.4) | **65.3** ($\pm$1.1) | **78.9** ($\pm$3.3) | **62.5** ($\pm$1.9) |

Table 28: `A/Type`. Results from Table 2 extended to each matrix configuration.

| | Mean | Center | 2x2Grid | 3x3Grid | L-R | U-D | O-IC | O-IG |
|---|---|---|---|---|---|---|---|---|
| ALANS | 19.0 ($\pm$3.4) | 18.5 ($\pm$4.9) | 19.7 ($\pm$1.7) | 18.9 ($\pm$3.8) | 22.1 ($\pm$3.5) | 20.8 ($\pm$4.5) | 17.6 ($\pm$3.1) | 15.2 ($\pm$3.1) |
| CPCNet | 38.6 ($\pm$4.3) | 33.3 ($\pm$4.5) | 45.5 ($\pm$5.0) | 42.4 ($\pm$3.1) | 40.8 ($\pm$3.4) | 40.4 ($\pm$4.3) | 32.2 ($\pm$5.0) | 34.7 ($\pm$6.6) |
| CNN-LSTM | 14.5 ($\pm$0.8) | 13.1 ($\pm$0.7) | 17.0 ($\pm$0.7) | 17.3 ($\pm$0.8) | 13.8 ($\pm$1.9) | 13.5 ($\pm$0.8) | 13.9 ($\pm$1.8) | 12.9 ($\pm$1.1) |
| CoPINet | 19.8 ($\pm$0.9) | 18.4 ($\pm$1.8) | 25.8 ($\pm$0.5) | 25.5 ($\pm$0.6) | 18.7 ($\pm$1.7) | 16.9 ($\pm$1.6) | 13.8 ($\pm$1.0) | 19.5 ($\pm$0.8) |
| DRNet | 44.3 ($\pm$0.8) | 38.7 ($\pm$1.8) | 52.0 ($\pm$1.5) | 50.2 ($\pm$2.4) | 41.6 ($\pm$0.4) | 41.4 ($\pm$0.8) | 38.7 ($\pm$1.7) | 47.3 ($\pm$0.3) |
| MRNet | 19.4 ($\pm$0.3) | 2.7 ($\pm$0.2) | 48.0 ($\pm$1.8) | 44.9 ($\pm$0.4) | 9.0 ($\pm$3.8) | 7.0 ($\pm$0.9) | 6.6 ($\pm$1.3) | 17.8 ($\pm$6.0) |
| PrAE | 37.0 ($\pm$1.7) | 37.1 ($\pm$1.8) | 46.7 ($\pm$1.6) | 30.9 ($\pm$1.8) | 46.6 ($\pm$2.1) | 46.9 ($\pm$2.5) | 25.6 ($\pm$4.7) | 25.4 ($\pm$0.6) |
| PredRNet | 40.2 ($\pm$1.3) | 30.3 ($\pm$2.3) | 49.1 ($\pm$1.2) | 51.8 ($\pm$2.4) | 39.6 ($\pm$2.5) | 37.5 ($\pm$2.2) | 32.8 ($\pm$2.2) | 39.9 ($\pm$0.8) |
| RelBase | 44.1 ($\pm$1.0) | 39.4 ($\pm$0.9) | 50.9 ($\pm$0.5) | 51.0 ($\pm$0.4) | 40.6 ($\pm$2.4) | 41.6 ($\pm$0.8) | 37.6 ($\pm$1.5) | 47.1 ($\pm$1.4) |
| SCL | 49.5 ($\pm$1.8) | 47.5 ($\pm$2.4) | 58.8 ($\pm$4.0) | 57.6 ($\pm$0.6) | 47.8 ($\pm$1.1) | 47.2 ($\pm$1.6) | 40.9 ($\pm$4.0) | 46.3 ($\pm$2.0) |
| SRAN | 30.7 ($\pm$2.2) | 30.2 ($\pm$1.0) | 37.3 ($\pm$1.3) | 36.3 ($\pm$0.9) | 33.0 ($\pm$3.3) | 32.3 ($\pm$3.5) | 21.3 ($\pm$4.6) | 23.9 ($\pm$1.9) |
| STSN | 39.1 ($\pm$5.0) | 48.5 ($\pm$3.5) | 44.6 ($\pm$11.3) | 40.6 ($\pm$9.7) | 38.3 ($\pm$8.5) | 36.8 ($\pm$7.7) | 35.1 ($\pm$1.6) | 30.1 ($\pm$0.9) |
| WReN | 15.1 ($\pm$0.7) | 15.5 ($\pm$2.0) | 15.6 ($\pm$0.5) | 16.3 ($\pm$1.9) | 14.1 ($\pm$0.1) | 14.8 ($\pm$0.3) | 14.5 ($\pm$1.4) | 15.0 ($\pm$0.5) |
| PoNG (ours) | **59.4** ($\pm$6.9) | **58.4** ($\pm$7.4) | **69.7** ($\pm$7.9) | **65.4** ($\pm$7.4) | **56.6** ($\pm$8.1) | **56.6** ($\pm$8.9) | **53.2** ($\pm$6.3) | **55.4** ($\pm$4.2) |

Table 29: A/ColorSize. The table presents results from Table 10 extended to each matrix configuration.

| | Mean | Center | 2x2-Grid | 3x3-Grid | L-R | U-D | O-IC | O-IG |
|---|---|---|---|---|---|---|---|---|
| ALANS | 15.1 (±3.3) | 15.2 (±4.3) | 17.2 (±2.7) | 17.2 (±3.9) | 14.9 (±3.4) | 13.8 (±4.5) | 13.7 (±1.9) | 13.7 (±2.4) |
| CPCNet | 33.0 (±5.3) | 33.0 (±4.4) | 34.6 (±4.7) | 32.6 (±2.2) | 28.0 (±5.7) | 30.8 (±6.5) | 35.8 (±7.6) | 36.3 (±7.8) |
| CNN-LSTM | 13.4 (±0.9) | 14.3 (±1.0) | 15.6 (±2.4) | 14.4 (±1.0) | 13.4 (±0.3) | 13.8 (±1.1) | 11.7 (±1.7) | 10.4 (±0.8) |
| CoPINet | 18.3 (±0.3) | 19.7 (±0.8) | 24.8 (±0.3) | 23.5 (±2.0) | 13.9 (±1.3) | 15.2 (±0.3) | 12.5 (±0.5) | 18.4 (±0.5) |
| DRNet | 38.3 (±0.5) | 38.3 (±0.9) | 44.2 (±2.9) | 39.5 (±2.2) | 33.0 (±1.1) | 35.2 (±3.3) | 37.2 (±3.6) | 40.2 (±3.1) |
| MRNet | 18.7 (±1.1) | 17.6 (±0.9) | 26.5 (±2.6) | 26.7 (±1.8) | 13.8 (±1.6) | 13.9 (±1.1) | 14.8 (±1.6) | 18.1 (±0.9) |
| PrAE | 30.0 (±1.1) | 27.1 (±2.6) | 40.1 (±1.2) | 30.8 (±1.9) | 29.9 (±1.3) | 30.6 (±1.2) | 26.2 (±1.5) | 25.7 (±0.5) |
| PredRNet | 31.0 (±1.6) | 31.2 (±2.4) | 40.5 (±0.8) | 37.0 (±2.3) | 25.9 (±1.6) | 27.9 (±2.9) | 23.7 (±0.5) | 30.0 (±1.4) |
| RelBase | 36.6 (±0.8) | 36.5 (±0.5) | 41.4 (±0.3) | 38.4 (±1.3) | 32.2 (±0.4) | **37.1** (±0.6) | 33.2 (±3.2) | 36.8 (±1.6) |
| SCL | 40.8 (±3.2) | 40.8 (±2.4) | 49.2 (±8.2) | 45.7 (±5.5) | 29.8 (±3.3) | 30.7 (±2.0) | 43.4 (±2.9) | **46.0** (±2.4) |
| SRAN | 22.7 (±1.1) | 18.7 (±2.1) | 29.3 (±1.8) | 26.6 (±0.5) | 19.0 (±2.4) | 18.4 (±2.5) | 22.5 (±1.9) | 23.7 (±0.8) |
| STSN | 27.3 (±4.6) | 28.9 (±6.1) | 30.3 (±5.3) | 29.3 (±5.9) | 24.3 (±6.0) | 24.6 (±5.0) | 28.8 (±5.8) | 26.1 (±1.4) |
| WReN | 13.5 (±0.1) | 14.6 (±0.7) | 14.3 (±0.2) | 15.0 (±0.4) | 15.2 (±0.2) | 14.4 (±0.8) | 11.1 (±1.2) | 10.1 (±0.2) |
| PoNG (ours) | **44.7** (±2.1) | **46.1** (±3.1) | **53.5** (±2.5) | **48.4** (±3.4) | **36.9** (±3.1) | **35.1** (±2.9) | **48.6** (±4.0) | 44.2 (±1.6) |

Table 30: A/ColorType. The table presents results from Table 10 extended to each matrix configuration.

| | Mean | Center | 2x2-Grid | 3x3-Grid | L-R | U-D | O-IC | O-IG |
|---|---|---|---|---|---|---|---|---|
| ALANS | 17.7 (±3.2) | 17.4 (±4.4) | 18.4 (±1.6) | 17.6 (±2.4) | 19.7 (±5.9) | 19.0 (±5.4) | 17.1 (±3.4) | 14.9 (±1.3) |
| CPCNet | 25.0 (±0.9) | 21.0 (±1.2) | 33.0 (±4.5) | 28.4 (±1.6) | 23.3 (±1.1) | 20.9 (±1.7) | 21.5 (±2.4) | 26.5 (±2.0) |
| CNN-LSTM | 14.7 (±1.7) | 13.9 (±3.2) | 18.6 (±1.6) | 16.7 (±1.9) | 14.7 (±1.3) | 12.8 (±0.8) | 13.4 (±3.0) | 12.4 (±2.8) |
| CoPINet | 17.2 (±0.1) | 14.4 (±0.5) | 23.2 (±0.8) | 23.6 (±0.7) | 13.6 (±0.7) | 13.5 (±0.6) | 13.2 (±1.0) | 18.6 (±1.0) |
| DRNet | 29.5 (±0.5) | 22.7 (±2.0) | 36.7 (±0.3) | 34.7 (±0.5) | 24.3 (±1.2) | 23.0 (±1.6) | 27.2 (±0.6) | **38.0** (±0.2) |
| MRNet | 20.0 (±2.6) | 15.5 (±0.6) | 31.3 (±2.7) | 28.6 (±0.7) | 15.0 (±3.9) | 14.6 (±4.4) | 15.1 (±3.6) | 19.7 (±3.7) |
| PrAE | 26.7 (±0.7) | 24.3 (±1.0) | 35.9 (±0.5) | 27.3 (±1.6) | **30.3** (±1.2) | 28.1 (±0.9) | 17.5 (±0.6) | 23.1 (±1.1) |
| PredRNet | 28.0 (±0.7) | 20.6 (±1.1) | 38.3 (±1.6) | 35.2 (±1.6) | 26.1 (±1.4) | 23.8 (±0.6) | 23.3 (±1.1) | 28.7 (±1.5) |
| RelBase | 29.7 (±0.6) | 23.2 (±0.8) | 36.6 (±0.6) | 34.4 (±1.0) | 25.0 (±2.0) | 23.7 (±0.9) | 27.5 (±1.1) | 37.0 (±0.2) |
| SCL | 32.0 (±2.3) | 25.9 (±3.8) | **44.4** (±2.4) | 40.1 (±1.9) | 26.1 (±3.0) | 24.4 (±1.8) | 27.1 (±1.9) | 35.7 (±2.1) |
| SRAN | 20.9 (±0.9) | 20.2 (±2.6) | 27.2 (±1.9) | 24.9 (±1.9) | 19.2 (±0.6) | 18.2 (±0.9) | 15.7 (±0.5) | 20.5 (±0.7) |
| STSN | 21.9 (±4.6) | 21.9 (±4.1) | 26.9 (±7.7) | 25.3 (±8.1) | 20.5 (±2.0) | 20.7 (±2.7) | 18.6 (±4.7) | 19.8 (±4.7) |
| WReN | 13.8 (±0.7) | 13.1 (±2.0) | 16.0 (±0.5) | 14.5 (±0.7) | 13.3 (±1.1) | 14.0 (±1.1) | 13.3 (±0.3) | 13.1 (±1.1) |
| PoNG (ours) | **34.3** (±0.8) | **29.4** (±1.4) | 43.4 (±1.4) | **41.2** (±2.8) | 29.5 (±1.2) | **29.2** (±0.8) | **29.7** (±1.6) | 37.2 (±1.6) |

Table 31: A/SizeType. The table presents results from Table 10 extended to each matrix configuration.

| | Mean | Center | 2x2-Grid | 3x3-Grid | L-R | U-D | O-IC | O-IG |
|---|---|---|---|---|---|---|---|---|
| ALANS | 15.7 (±3.2) | 14.0 (±2.0) | 18.6 (±5.4) | 18.2 (±5.7) | 15.1 (±2.0) | 14.8 (±1.8) | 14.0 (±1.3) | 15.6 (±5.3) |
| CPCNet | 24.1 (±1.2) | 26.3 (±1.8) | 30.1 (±2.0) | 29.0 (±0.4) | 22.3 (±0.6) | 22.8 (±0.5) | 17.8 (±1.4) | 20.1 (±2.5) |
| CNN-LSTM | 13.0 (±0.1) | 13.1 (±0.4) | 14.2 (±0.9) | 14.0 (±0.6) | 13.2 (±0.9) | 12.1 (±0.0) | 12.3 (±0.7) | 12.6 (±1.2) |
| CoPINet | 19.7 (±0.7) | 20.9 (±1.6) | 23.5 (±0.8) | 22.6 (±0.9) | 19.1 (±1.1) | 19.0 (±0.7) | 17.4 (±1.8) | 15.5 (±0.9) |
| DRNet | 31.6 (±1.2) | 35.6 (±2.6) | 40.4 (±0.7) | 37.1 (±1.1) | 28.4 (±3.1) | 27.5 (±1.7) | 22.9 (±0.7) | **29.4** (±1.5) |
| MRNet | 28.2 (±0.9) | 31.7 (±2.0) | 34.9 (±2.4) | 31.0 (±1.5) | 27.2 (±3.5) | 27.1 (±3.3) | 21.1 (±1.3) | 24.4 (±2.1) |
| PrAE | 25.6 (±0.8) | 23.1 (±1.1) | 35.2 (±1.0) | 27.1 (±2.0) | 28.0 (±0.6) | 26.3 (±0.9) | 17.6 (±0.3) | 21.9 (±1.3) |
| PredRNet | 27.9 (±0.5) | 25.8 (±1.3) | 38.4 (±1.3) | 35.2 (±0.6) | 25.0 (±0.7) | 26.7 (±0.7) | 17.8 (±1.5) | 25.9 (±0.9) |
| RelBase | 31.1 (±1.0) | 33.3 (±3.0) | 39.3 (±0.8) | 37.9 (±0.8) | 28.4 (±1.9) | 29.2 (±1.9) | 22.1 (±1.6) | 27.5 (±1.2) |
| SCL | **33.5** (±0.7) | **41.3** (±0.6) | **44.0** (±2.4) | 39.7 (±0.8) | **30.7** (±0.4) | **29.3** (±1.6) | **23.2** (±0.6) | 26.1 (±0.4) |
| SRAN | 23.3 (±0.3) | 23.7 (±2.6) | 30.8 (±1.7) | 28.2 (±0.6) | 21.0 (±2.6) | 22.7 (±0.8) | 16.0 (±0.6) | 20.6 (±0.7) |
| STSN | 12.3 (±0.1) | 12.9 (±1.0) | 12.0 (±0.8) | 12.4 (±1.8) | 11.9 (±0.2) | 11.9 (±0.7) | 12.2 (±0.6) | 12.3 (±0.6) |
| WReN | 14.1 (±0.2) | 14.3 (±0.5) | 15.2 (±0.8) | 14.7 (±0.4) | 14.0 (±0.8) | 14.7 (±0.5) | 12.1 (±0.4) | 13.6 (±0.7) |
| PoNG (ours) | 32.1 (±2.1) | 34.6 (±3.3) | 42.4 (±0.7) | **39.9** (±2.8) | 28.8 (±1.8) | 28.4 (±1.9) | 22.1 (±4.4) | 28.6 (±2.4) |

Table 32: `A/Color-Progression`. The table presents results from Table 10 extended to each matrix configuration.

| | Mean | Center | 2x2-Grid | 3x3-Grid | L-R | U-D | O-IC | O-IG |
|---|---|---|---|---|---|---|---|---|
| ALANS | 24.8 (±18.8) | 24.9 (±21.3) | 24.5 (±13.8) | 24.3 (±16.4) | 26.3 (±23.3) | 29.3 (±26.3) | 24.5 (±20.1) | 19.7 (±10.1) |
| CPCNet | 50.5 (±0.6) | 51.9 (±0.3) | 37.6 (±2.6) | 33.9 (±1.5) | 59.0 (±0.6) | 58.9 (±1.5) | 67.7 (±2.4) | 44.8 (±0.8) |
| CNN-LSTM | 17.2 (±1.5) | 20.3 (±2.2) | 17.7 (±1.4) | 16.9 (±2.4) | 16.0 (±1.3) | 15.6 (±1.4) | 18.0 (±1.0) | 15.7 (±1.9) |
| CoPINet | 35.8 (±0.6) | 37.3 (±0.9) | 29.8 (±1.8) | 28.0 (±0.3) | 35.8 (±0.5) | 35.8 (±0.9) | 44.2 (±0.8) | 40.1 (±2.5) |
| DRNet | 72.8 (±1.3) | 71.3 (±2.5) | 71.9 (±1.8) | 65.5 (±0.7) | 66.8 (±1.8) | 70.4 (±2.7) | 85.4 (±0.7) | 77.6 (±1.4) |
| MRNet | 34.4 (±3.4) | 45.6 (±6.1) | 33.1 (±8.9) | 30.1 (±9.1) | 24.1 (±1.8) | 23.7 (±4.5) | 49.6 (±1.5) | 35.2 (±1.1) |
| PrAE | 62.3 (±0.9) | 73.3 (±2.0) | 75.0 (±3.3) | 41.4 (±5.9) | **83.0** (±0.7) | **82.9** (±1.2) | 47.1 (±4.7) | 33.0 (±1.7) |
| PredRNet | 62.3 (±2.2) | 64.2 (±1.2) | 61.6 (±5.8) | 52.0 (±5.2) | 64.3 (±1.5) | 64.1 (±0.8) | 75.3 (±0.9) | 54.8 (±6.1) |
| RelBase | 73.0 (±1.8) | 73.8 (±4.0) | 72.4 (±0.6) | 65.1 (±0.2) | 69.7 (±2.9) | 73.0 (±5.1) | 83.1 (±2.7) | 73.8 (±2.5) |
| SCL | 75.6 (±10.1) | 84.0 (±6.3) | 74.8 (±10.3) | 66.4 (±14.7) | 73.2 (±13.1) | 71.6 (±12.4) | 81.1 (±7.5) | 77.6 (±8.8) |
| SRAN | 42.1 (±2.3) | 47.7 (±4.3) | 35.5 (±0.9) | 31.4 (±0.7) | 41.8 (±4.4) | 42.2 (±4.1) | 51.3 (±1.9) | 44.6 (±1.2) |
| STSN | 39.9 (±14.7) | 47.1 (±9.6) | 35.8 (±11.4) | 32.3 (±10.5) | 38.5 (±14.2) | 39.5 (±14.2) | 49.1 (±26.5) | 38.1 (±18.6) |
| WReN | 18.0 (±0.4) | 20.7 (±1.0) | 18.8 (±0.3) | 17.4 (±0.7) | 15.8 (±0.6) | 15.1 (±0.5) | 19.3 (±0.7) | 18.8 (±0.4) |
| PoNG (ours) | **81.4** (±3.1) | **88.3** (±6.2) | **86.5** (±5.0) | **79.8** (±3.3) | 73.4 (±3.0) | 71.1 (±2.1) | **87.2** (±2.8) | **83.5** (±1.0) |

Table 33: `A/Color-Arithmetic`. The table presents results from Table 10 extended to each matrix configuration.

| | Mean | Center | 2x2-Grid | 3x3-Grid | L-R | U-D | O-IC | O-IG |
|---|---|---|---|---|---|---|---|---|
| ALANS | 18.3 (±6.6) | 18.6 (±6.4) | 18.2 (±5.5) | 17.0 (±5.0) | 20.0 (±8.2) | 18.8 (±7.6) | 18.1 (±8.0) | 17.4 (±5.7) |
| CPCNet | 45.9 (±2.7) | 41.2 (±3.3) | 39.2 (±5.7) | 35.6 (±5.2) | 50.5 (±1.0) | 47.7 (±0.8) | 62.2 (±1.7) | 45.5 (±2.7) |
| CNN-LSTM | 17.1 (±3.7) | 18.8 (±5.9) | 17.3 (±4.2) | 17.3 (±3.4) | 15.8 (±3.2) | 15.6 (±2.4) | 17.2 (±3.9) | 17.8 (±3.6) |
| CoPINet | 35.2 (±0.5) | 35.9 (±0.5) | 31.4 (±0.5) | 27.7 (±0.7) | 34.1 (±1.2) | 33.5 (±1.6) | 43.4 (±0.7) | 40.3 (±0.5) |
| DRNet | 66.7 (±1.2) | 60.0 (±2.2) | 69.1 (±1.8) | 62.5 (±1.9) | 63.3 (±2.5) | **63.2** (±4.2) | **77.5** (±3.0) | 71.6 (±2.2) |
| MRNet | 35.7 (±5.9) | 38.9 (±14.1) | 38.8 (±3.1) | 32.4 (±3.2) | 20.6 (±7.5) | 20.4 (±6.9) | 46.6 (±7.0) | 52.4 (±0.8) |
| PrAE | 43.0 (±26.5) | 47.2 (±30.6) | 52.5 (±34.9) | 32.0 (±16.5) | 54.6 (±36.0) | 53.4 (±35.3) | 34.4 (±18.8) | 26.8 (±13.8) |
| PredRNet | 56.9 (±1.4) | 50.8 (±2.2) | 62.5 (±1.8) | 54.5 (±1.9) | 55.3 (±0.4) | 53.7 (±1.5) | 68.8 (±0.2) | 53.4 (±6.0) |
| RelBase | 66.2 (±1.0) | 60.4 (±1.6) | 69.4 (±0.4) | 61.9 (±3.0) | 63.1 (±1.1) | 62.4 (±0.4) | 74.4 (±1.9) | 72.0 (±1.8) |
| SCL | 60.0 (±4.1) | 62.8 (±5.4) | 58.5 (±2.9) | 54.0 (±2.2) | 54.6 (±6.6) | 52.3 (±5.8) | 69.1 (±5.2) | 68.8 (±1.4) |
| SRAN | 39.9 (±2.7) | 39.8 (±6.0) | 37.6 (±2.7) | 32.0 (±2.1) | 39.8 (±3.1) | 38.6 (±4.0) | 49.3 (±1.3) | 42.6 (±1.3) |
| STSN | 25.7 (±10.6) | 28.5 (±7.2) | 26.1 (±8.9) | 23.3 (±8.7) | 22.7 (±11.0) | 23.5 (±10.3) | 30.4 (±17.4) | 25.9 (±11.5) |
| WReN | 17.1 (±0.2) | 19.1 (±0.8) | 16.6 (±0.6) | 15.3 (±0.2) | 16.1 (±0.8) | 16.0 (±0.4) | 18.7 (±0.7) | 18.3 (±0.6) |
| PoNG (ours) | **70.0** (±4.1) | **64.4** (±7.9) | **74.4** (±3.9) | **69.8** (±3.2) | **63.4** (±3.3) | 62.1 (±5.1) | 76.0 (±4.5) | **80.1** (±2.1) |

Table 34: `A/Color-DistributeThree`. The table presents results from Table 10 extended to each matrix configuration.

| | Mean | Center | 2x2-Grid | 3x3-Grid | L-R | U-D | O-IC | O-IG |
|---|---|---|---|---|---|---|---|---|
| ALANS | 22.4 (±7.7) | 20.6 (±7.3) | 22.0 (±5.2) | 23.2 (±8.1) | 23.8 (±9.7) | 24.2 (±9.8) | 23.9 (±9.8) | 19.0 (±4.3) |
| CPCNet | 37.8 (±0.9) | 31.0 (±1.8) | 28.7 (±2.9) | 28.6 (±2.4) | 41.7 (±0.0) | 41.0 (±1.3) | 54.0 (±0.8) | 39.9 (±0.3) |
| CNN-LSTM | 20.6 (±6.7) | 24.6 (±10.1) | 20.1 (±4.9) | 19.2 (±5.2) | 18.7 (±5.3) | 17.9 (±4.1) | 22.5 (±9.2) | 21.7 (±7.9) |
| CoPINet | 26.9 (±0.5) | 25.8 (±1.0) | 24.1 (±0.1) | 21.8 (±0.7) | 22.9 (±1.6) | 23.4 (±0.2) | 34.4 (±1.2) | 35.6 (±1.0) |
| DRNet | 63.2 (±0.3) | 57.1 (±2.0) | 64.1 (±1.7) | 58.2 (±1.8) | 61.1 (±0.9) | 61.2 (±2.4) | 72.9 (±2.0) | 67.9 (±1.6) |
| MRNet | 18.6 (±0.1) | 20.9 (±3.4) | 27.7 (±3.3) | 24.0 (±6.6) | 11.9 (±6.1) | 13.2 (±2.5) | 14.3 (±4.6) | 18.3 (±6.6) |
| PrAE | 55.1 (±0.8) | 61.9 (±1.4) | 63.9 (±2.7) | 41.8 (±1.5) | 69.6 (±0.9) | 69.5 (±0.9) | 45.7 (±2.1) | 33.5 (±3.9) |
| PredRNet | 48.5 (±0.9) | 38.7 (±0.5) | 51.2 (±3.8) | 43.7 (±4.5) | 46.2 (±0.3) | 45.8 (±0.3) | 61.0 (±0.8) | 52.7 (±3.4) |
| RelBase | 65.7 (±4.6) | 60.8 (±6.2) | 64.9 (±3.8) | 57.9 (±5.5) | 67.6 (±5.3) | 66.1 (±4.8) | 75.6 (±2.8) | 67.2 (±4.8) |
| SCL | 63.9 (±4.3) | 70.8 (±4.5) | 60.3 (±4.3) | 54.5 (±3.7) | 59.7 (±5.8) | 57.8 (±6.4) | 72.0 (±4.0) | 71.8 (±2.2) |
| SRAN | 34.6 (±3.6) | 38.4 (±8.4) | 32.5 (±4.5) | 30.1 (±2.8) | 32.6 (±3.1) | 31.5 (±3.1) | 40.8 (±2.3) | 36.6 (±2.2) |
| STSN | 20.7 (±7.7) | 20.8 (±6.4) | 19.0 (±6.8) | 19.1 (±7.7) | 18.2 (±6.5) | 16.1 (±4.7) | 27.5 (±14.1) | 25.0 (±10.6) |
| WReN | 17.7 (±0.6) | 21.0 (±0.8) | 17.3 (±0.5) | 16.4 (±0.4) | 16.0 (±0.7) | 16.3 (±1.1) | 18.0 (±1.7) | 19.1 (±2.1) |
| PoNG (ours) | **81.3** (±1.6) | **84.1** (±4.9) | **82.7** (±2.1) | **77.6** (±1.4) | **80.6** (±2.8) | **80.9** (±3.5) | **84.4** (±2.1) | **79.1** (±3.5) |

## F  DATASHEETS FOR DATASETS

In what follows, we provide the description of the introduced datasets following the Datasheets for Datasets template (Gebru et al., 2021).

---

### Motivation

**For what purpose was the dataset created?** Was there a specific task in mind? Was there a specific gap that needed to be filled? Please provide a description.

The datasets were created to study generalization and knowledge transfer abilities of AVR models.

**Who created this dataset (e.g., which team, research group) and on behalf of which entity (e.g., company, institution, organization)?**

Hidden for blind review.

**Who funded the creation of the dataset?** If there is an associated grant, please provide the name of the grantor and the grant name and number.

Hidden for blind review.

**Any other comments?**

None.

---

### Composition

**What do the instances that comprise the dataset represent (e.g., documents, photos, people, countries)?** Are there multiple types of instances (e.g., movies, users, and ratings; people and interactions between them; nodes and edges)? Please provide a description.

Each dataset instance represents a single Raven's Progressive Matrix, which is a typical task used in human IQ tests.

**How many instances are there in total (of each type, if appropriate)?**

Each regime in Attributeless-I-RAVEN as well as the I-RAVEN-Mesh dataset contains $70\,000$ instances. The training, validation, and test splits contain $42\,000$, $14\,000$, and $14\,000$ matrices, resp. All together there are $770\,000$ $(11 \times 70\,000)$ instances.

**Does the dataset contain all possible instances or is it a sample (not necessarily random) of instances from a larger set?** If

the dataset is a sample, then what is the larger set? Is the sample representative of the larger set (e.g., geographic coverage)? If so, please describe how this representativeness was validated/verified. If it is not representative of the larger set, please describe why not (e.g., to cover a more diverse range of instances, because instances were withheld or unavailable).

The datasets contain a fixed number of instances generated with the data generator. Using a fixed seed ensures reproducibility of the generation process. The data generator allows to configure the number of generated samples.

**What data does each instance consist of? "Raw" data (e.g., unprocessed text or images) or features?** In either case, please provide a description.

Each RPM instance comprises 16 images that represent the RPM panels, a corresponding index of the correct answer and a representation of rules that govern the matrix. Section 3 of the paper provides additional details. Each instance is packaged as a separate file in the NPZ format, which is a widely-used binary format to store compressed NumPy arrays.

**Is there a label or target associated with each instance?** If so, please provide a description.

See above.

**Is any information missing from individual instances?** If so, please provide a description, explaining why this information is missing (e.g., because it was unavailable). This does not include intentionally removed information, but might include, e.g., redacted text.

There's no missing data.

**Are relationships between individual instances made explicit (e.g., users' movie ratings, social network links)?** If so, please describe how these relationships are made explicit.

There are no relationships between individual instances.

**Are there recommended data splits (e.g., training, development/validation, testing)?** If so, please provide a description of these splits, explaining the rationale behind them.

The datasets are split into training, validation and test splits. Each generated instance contains the split name in its filename, e.g., the `RAVEN_1_train.npz` file belongs to the train split.

**Are there any errors, sources of noise, or redundancies in the dataset?** If so, please provide a description.

To the best of our knowledge there are no errors, sources of noise, nor redundancies in the datasets.

**Is the dataset self-contained, or does it link to or otherwise rely on external resources (e.g., websites, tweets, other datasets)?** If it links to or relies on external resources, a) are there guarantees that they will exist, and remain constant, over time; b) are there official archival versions of the complete dataset (i.e., including the external resources as they existed at the time the dataset was created); c) are there any restrictions (e.g., licenses, fees) associated with any of the external resources that might apply to a future user? Please provide descriptions of all external resources and any restrictions associated with them, as well as links or other access points, as appropriate.

Both datasets are self-contained.

**Does the dataset contain data that might be considered confidential (e.g., data that is protected by legal privilege or by doctor-patient confidentiality, data that includes the content of individuals non-public communications)?** If so, please provide a description.

No.

**Does the dataset contain data that, if viewed directly, might be offensive, insulting, threatening, or might otherwise cause anxiety?** If so, please describe why.

No.

**Does the dataset relate to people?** If not, you may skip the remaining questions in this section.

No.

**Does the dataset identify any subpopulations (e.g., by age, gender)?** If so, please describe how these subpopulations are identified and provide a description of their respective distributions within the dataset.

N/A.

**Is it possible to identify individuals (i.e., one or more natural persons), either directly or indirectly (i.e., in combination with other data) from the dataset?** If so, please describe how.

N/A.

**Does the dataset contain data that might be considered sensitive in any way (e.g., data that reveals racial or ethnic origins, sexual orientations, religious beliefs, political opinions or union memberships, or locations; financial or health data; biometric or genetic data; forms of government identification, such as social security numbers; criminal history)?** If so, please provide a description.

N/A.

**Any other comments?**

None.

---

## Collection Process

**How was the data associated with each instance acquired?** Was the data directly observable (e.g., raw text, movie ratings), reported by subjects (e.g., survey responses), or indirectly inferred/derived from other data (e.g., part-of-speech tags, model-based guesses for age or language)? If data was reported by subjects or indirectly inferred/derived from other data, was the data validated/verified? If so, please describe how.

The data was generated with a computer program.

**What mechanisms or procedures were used to collect the data (e.g., hardware apparatus or sensor, manual human curation, software program, software API)?** How were these mechanisms or procedures validated?

We extended the data generation code used to create I-RAVEN: `https://github.com/husheng12345/SRAN`. A subset of the dataset was reviewed manually to ensure correctness of the generated matrices.

**If the dataset is a sample from a larger set, what was the sampling strategy (e.g., deterministic, probabilistic with specific sampling probabilities)?**

The dataset is produced by a generator that creates new RPM instances subject to specified constraints through a pseudo-random process. We use a fixed seed to ensure reproducibility of the generation process.

**Who was involved in the data collection process (e.g., students, crowdworkers, contractors) and how were they compensated (e.g., how much were crowdworkers paid)?**

The data generator has been written by the authors of this paper without delegating the work to other individuals.

**Over what timeframe was the data collected? Does this timeframe match the creation timeframe of the data associated with the instances (e.g., recent crawl of old news articles)?** If not, please describe the timeframe in which the data associated with the instances was created.

Development of the datasets lasted from January 2022 to September 2024.

**Were any ethical review processes conducted (e.g., by an institutional review board)?** If so, please provide a description of these review processes, including the outcomes, as well as a link or other access point to any supporting documentation.

N/A.

**Does the dataset relate to people?** If not, you may skip the remaining questions in this section.

No.

**Did you collect the data from the individuals in question directly, or obtain it via third parties or other sources (e.g., websites)?**

N/A.

**Were the individuals in question notified about the data collection?** If so, please describe (or show with screenshots or other information) how notice was provided, and provide a link or other access point to, or otherwise reproduce, the exact language of the notification itself.

N/A.

**Did the individuals in question consent to the collection and use of their data?** If so, please describe (or show with screenshots or other information) how consent was requested and provided, and provide a link or other access point to, or otherwise reproduce, the exact language to which the individuals consented.

N/A.

**If consent was obtained, were the consenting individuals provided with a mechanism to revoke their consent in the future or for certain uses?** If so, please provide a description, as well as a link or other access point to the mechanism (if appropriate).

N/A.

**Has an analysis of the potential impact of the dataset and its use on data subjects (e.g., a data protection impact analysis) been conducted?** If so, please provide a description of this analysis, including the outcomes, as well as a link or other access point to any supporting documentation.

N/A.

**Any other comments?**

We bear all responsibility in case of violation of rights.

## Preprocessing/cleaning/labeling

**Was any preprocessing/cleaning/labeling of the data done (e.g., discretization or bucketing, tokenization, part-of-speech tagging, SIFT feature extraction, removal of instances, processing of missing values)?** If so, please provide a description. If not, you may skip the remainder of the questions in this section.

The data created by the generator is ready to be used in a model. No preprocessing, cleaning, or labeling is required.

**Was the "raw" data saved in addition to the preprocessed/cleaned/labeled data (e.g., to support unanticipated future uses)?** If so, please provide a link or other access point to the "raw" data.

N/A.

**Is the software used to preprocess/clean/label the instances available?** If so, please provide a link or other access point.

No specific software is required to preprocess, clean, or label the instances. The released code repository contains the code required to reproduce all experiments from the paper, which can be used as a reference implementation for loading the datasets.

**Any other comments?**

None.

## Uses

**Has the dataset been used for any tasks already?** If so, please provide a description.

The datasets have been used to conduct experiments presented in the paper.

**Is there a repository that links to any or all papers or systems that use the dataset?** If so, please provide a link or other access point.

N/A.

**What (other) tasks could the dataset be used for?**

The datasets could be used in a multi-task setting to improve abstract reasoning capabilities of computer vision models.

**Is there anything about the composition of the dataset or the way it was collected and pre-processed/cleaned/labeled that might impact future uses?** For example, is there anything that a future user might need to know to avoid uses that could result in unfair treatment of individuals or groups (e.g., stereotyping, quality of service issues) or other undesirable harms (e.g., financial harms, legal risks) If so, please provide a description. Is there anything a future user could do to mitigate these undesirable harms?

We do not see any undesirable harms that could apply to future users of the datasets.

**Are there tasks for which the dataset should not be used?** If so, please provide a description.

The datasets should not be used in human IQ tests, as they were explicitly designed to assess generalization and knowledge transfer abilities of deep learning models.

**Any other comments?**

None.

---

### Distribution

---

**Will the dataset be distributed to third parties outside of the entity (e.g., company, institution, organization) on behalf of which the dataset was created?** If so, please provide a description.

The datasets will become publicly available after paper acceptance. Additionally, as discussed in Section 4 ("Reproducibility"), the attached code allows for generation of all datasets from scratch, eliminating the dependency on file-hosting services required to distribute the data.

**How will the dataset will be distributed (e.g., tarball on website, API, GitHub)** Does the dataset have a digital object identifier (DOI)?

The datasets will be shared on GitHub.

**When will the dataset be distributed?**

The datasets will become publicly available after paper acceptance.

**Will the dataset be distributed under a copyright or other intellectual property (IP) license, and/or under applicable terms of use (ToU)?** If so, please describe this license and/or ToU, and provide a link or other access point to, or otherwise reproduce, any relevant licensing terms or ToU, as well as any fees associated with these restrictions.

The code repository is released under the GPL-3.0 license. This follows the license associated with the generators of the base datasets – RAVEN (`https://github.com/WellyZhang/RAVEN`) and I-RAVEN (`https://github.com/husheng12345/SRAN`). The datasets introduced in this paper are released under the CC license.

**Have any third parties imposed IP-based or other restrictions on the data associated with the instances?** If so, please describe these restrictions, and provide a link or other access point to, or otherwise reproduce, any relevant licensing terms, as well as any fees associated with these restrictions.

No.

**Do any export controls or other regulatory restrictions apply to the dataset or to individual instances?** If so, please describe these restrictions, and provide a link or other access point to, or otherwise reproduce, any supporting documentation.

No.

**Any other comments?**

None.

---

### Maintenance

---

**Who will be supporting/hosting/maintaining the dataset?**

Hidden for blind review.

**How can the owner/curator/manager of the dataset be contacted (e.g., email address)?**

Hidden for blind review.

**Is there an erratum?** If so, please provide a link or other access point.

No.

**Will the dataset be updated (e.g., to correct labeling errors, add new instances, delete instances)?** If so, please describe how often, by whom, and how updates will be communicated to users (e.g., mailing list, GitHub)?

Future changes will be documented in release notes in the code repository.

**If the dataset relates to people, are there applicable limits on the retention of the data associated with the instances (e.g., were individuals in question told that their data would be retained for a fixed period of time and then deleted)?** If so, please describe these limits and explain how they will be enforced.

N/A.

**Will older versions of the dataset continue to be supported/hosted/maintained?** If so, please describe how. If not, please describe how its obsolescence will be communicated to users.

Older versions of the dataset will be available in the history of the code repository.

**If others want to extend/augment/build on/contribute to the dataset, is there a mechanism for them to do so?** If so, please provide a description. Will these contributions be validated/verified? If so, please describe how. If not, why not? Is there a process for communicating/distributing these contributions to other users? If so, please provide a description.

Contributions are welcome. GitHub Issues of the code repository will be used to communicate between contributors and project maintainers.

**Any other comments?**

None.

