# OpenReview forum: "Generalization and Knowledge Transfer in Abstract Visual Reasoning Models"
_ICLR.cc/2025/Conference — ICLR 2025 Conference Withdrawn Submission_

### Official Review · Reviewer_iGdB · 2024-11-03

**Soundness:** 3
**Presentation:** 3
**Contribution:** 2
**Rating:** 3
**Confidence:** 3

**Summary:**

The authors investigate the generalization and knowledge reuse capabilities of deep neural networks in abstract visual reasoning (AVR) using Raven's Progressive Matrices (RPMs). They introduce two novel benchmarks, Attributeless-I-RAVEN and I-RAVEN-Mesh, aimed at assessing the generalization of abstract rules and supporting progressive knowledge acquisition in transfer learning contexts. Furthermore, the proposed Pathways of Normalized Group Convolution (PoNG) model shows superior performance on these benchmarks as well as in visual analogy tasks, indicating its effectiveness in AVR beyond RPMs.

**Strengths:**

This work proposes a novel suite of generalization challenges stemming from I-RAVEN, the authors introduce Attributeless-I-RAVEN, comprising 10 generalization regime, and I-RAVEN-Mesh, a variant of I-RAVEN with a new grid-like structure overlaid on the matrices.
Finally, they propose Pathways of Normalized Group Convolution (PoNG), a novel AVR model. The paper is well-presented.

**Weaknesses:**

In this paper, the authors claim, "Compared to PGM, our datasets feature compositionality and variety of figure configurations, and their processing doesn’t require substantial computational resources." While this addresses some issues with PGM to a certain extent, it also constitutes a form of redundant work, and the newly generated Raven puzzles do not appear to be more difficult than those in PGM. Additionally, the proposed PoNG does not seem particularly novel.

**Questions:**

**Q1:** The parallel pathway proposed by PoNG shares similar concepts with DRNet (2024), which limits the novelty of PoNG.

**Q2:** The performance of PoNG also does not appear to surpass that of DRNet (2024); the authors should provide a comparison with DRNet (2024).

**Q3:** The authors should restate the significance of their proposed Attributeless-I-RAVEN and I-RAVEN-Mesh datasets.


Ref: Zhao, K., Xu, C., & Si, B. (2024, March). Learning Visual Abstract Reasoning through Dual-Stream Networks. In Proceedings of the AAAI Conference on Artificial Intelligence (Vol. 38, No. 15, pp. 16979-16988).

---

> ### Author Response · Authors · 2024-11-28
> **Response to Reviewer iGdB**
>
> > Q1: The parallel pathway proposed by PoNG shares similar concepts with DRNet (2024), which limits the novelty of PoNG.
>
> Thank you for bringing up this relevant reference. We have added DRNet to the revised paper, referencing it in the “Model architectures” paragraph in Section 2 and including it in our set of evaluated models across all experiments. Both PoNG and DRNet are related to RelBase and we explicitly acknowledged this connection in our paper, e.g. in the “Panel encoder” paragraph in Section 3: “Following RelBase (Spratley et al., 2020), [...]”.
>
> DRNet builds upon RelBase by employing two panel encoders, one derived from RelBase and another utilizing a Vision Transformer. It also introduces the Integration Module that combines these two panel representations via tensor concatenation and a linear projection.
>
> PoNG also builds on RelBase’s panel encoder, incorporating adjustments as described in the “Panel encoder” paragraph in Section 3: “Differently from RelBase, [...]”. The primary novelty of PoNG lies in its panel embedding aggregation module, see the “Reasoner” paragraph in Section 3. In particular, we introduce the Pathways block, as described in the “Pathways” paragraph in Section 3. The block features the group (P3) and group-pair (P4) convolution modules, which are novel contributions not explored in prior work. Additionally, we propose the “target-conditioned rule head” as detailed in the “Answer prediction” paragraph in Section 3.
>
> While both DRNet and PoNG build upon RelBase, their contributions are orthogonal. Combining these complementary advancements in a joint architecture opens a promising direction for future work.
>
> > Q2: The performance of PoNG also does not appear to surpass that of DRNet (2024); the authors should provide a comparison with DRNet (2024).
>
> In the revision, we have included DRNet in the set of evaluated models across all experiments. DRNet reports 97.6% accuracy on I-RAVEN, exceeding the performance of PoNG (95.9%) by 1.7 p.p. With the experimental setup employed in our work (see “Experimental setup” in Section 4), DRNet demonstrates strong performance, achieving the 2nd best result on I-RAVEN, A/Color, A/Position (Table 2), and A/Color-Arithmetic (Table 10). However, PoNG outperforms DRNet in all A-I-RAVEN regimes. Experments regarding progressive knowledge acquisition on I-RAVEN-Mesh show that DRNet presents strong transfer learning capacity (Fig. 6).
>
> In addition, we have also included results for MRNet, ARII, DRNet, and Slot-Abstractor in Table 4, which presents results on PGM. On average, DRNet achieves the highest performance (58.3%), narrowly surpassing PoNG (57.3%) by 1 p.p. PoNG ranks 2nd in the Neutral, Held-out Attribute Pairs, and Held-out Triples regimes, and achieves the best performance in the Held-out Triple Pairs regime.
>
> Based on this updated analysis, PoNG excels in the introduced generalization challenges, while DRNet performs better on PGM. This disparity may relate to the size of the models – PoNG, with 3.1M parameters, is smaller than DRNet, which contains 24.7M parameters. In result, PoNG may learn more efficiently from the relatively smaller datasets in the RAVEN line, while DRNet’s larger capacity may help to better utilize the larger PGM dataset.
>
> > Q3: The authors should restate the significance of their proposed Attributeless-I-RAVEN and I-RAVEN-Mesh datasets.
>
> Building on the broad adoption of RAVEN and the generalization evaluation capabilities of PGM, our datasets extend I-RAVEN to enable direct assessment of generalization and knowledge transfer in AVR models. In contrast to PGM, our datasets feature compositionality, a variety of figure configurations, and do not require substantial computational resources. Additionally, the datasets include structural annotations, which have been proven valuable in recent neuro-symbolic approaches.
>
> The moderate size of these datasets allowed us to evaluate 14 models across 11 diverse generalization challenges, a study that would not have been feasible with PGM using reasonable resources. For instance, each RAVEN-based dataset contains 42K training samples, which is approximately 28.6x fewer than the 1.2M training samples in PGM. This increased accessibility makes our benchmarks particularly useful for resource-constrained research and enables a more practical evaluation of AVR models focused on generalization.
>
> Moreover, without tuning PoNG specifically for PGM, we showed that the performance improvements on the introduced datasets transferred to PGM, showcasing the relevance of our benchmarks. A possible future work emerging from this improved accessibility may involve conducting a neural architecture search aimed at discovering new neural modules excelling in generalization.

---

> > ### Comment · Reviewer_iGdB · 2024-12-02
> >
> > I appreciate the author's response. Considering that PoNG uses meta-information to assist with training, I will maintain my score.

---

> > > ### Author Response · Authors · 2024-12-03
> > > **Response to Reviewer iGdB**
> > >
> > > We thank the Reviewer for their feedback and acknowledge that the use of meta-information is an important aspect to account for in evaluating model performance. PoNG’s reliance on such meta-information was explicitly noted in the ablation study: *"[..] signifying high relevance of the auxiliary training signal in PoNG’s training."*. However, we’d like to emphasize that PoNG represents only one out of four contributions outlined in Section 1. In addition to PoNG:
> > > 1. We introduce the Attributeless-I-RAVEN (A-I-RAVEN) dataset that enables measuring generalization across 10 regimes.
> > > 1. We construct I-RAVEN-Mesh, an extension of I-RAVEN with a new component structure that facilitates assessment of progressive knowledge acquisition in a TL setting.
> > > 1. We evaluate the performance of state-of-the-art AVR models on the introduced benchmarks, uncovering their limitations in terms of generalization to novel problem settings.
> > >
> > > We hope the above-mentioned contributions constitute an important addition to existing state-of-the-art research in the field of AVR and are valuable, even if the Reviewer does not rate PoNG as a major achievement.

---

### Official Review · Reviewer_7CrM · 2024-11-04

**Soundness:** 2
**Presentation:** 3
**Contribution:** 2
**Rating:** 3
**Confidence:** 4

**Summary:**

This paper investigates the generalization and knowledge transfer capabilities of deep neural networks in the domain of abstract visual reasoning (AVR), specifically focusing on Raven’s Progressive Matrices (RPMs). The authors introduce two new datasets, Attributeless-I-RAVEN and I-RAVEN-Mesh, and propose a new neural architecture called Pathways of Normalized Group Convolution (PoNG). The paper also takes a series of experiments to analysis different AVR models in these two datasets.

**Strengths:**

Pros:

1. The introduction of Attributeless-I-RAVEN with 10 generalization regimes is a contribution. It allows for a more comprehensive assessment of a model's ability to generalize to novel attributes and rule combinations, which is a crucial aspect of AVR. The detailed description of the regimes and the corresponding held-out attributes and rules provides a rich set of challenges for evaluating AVR models.

2. I-RAVEN-Mesh, with a grid-like structure embedded, is another valuable addition. It enables the study of progressive knowledge acquisition in a transfer learning setting, which is an important research direction in the field. The ability to overlay line-based patterns on the RPM matrices and the clear definition of the mesh component's attributes and rules add complexity and realism to the task.

3. This study presents a novel architecture that combines parallel design, weight sharing, and normalization techniques. The pathways block, with its four parallel pathways and the use of normalized group convolution operators, is a unique feature of the model.

4. The authors conduct an extensive evaluation on the proposed benchmarks and other related datasets. This comparative analysis provides valuable insights into the performance and limitations of existing models in terms of generalization and knowledge transfer.

**Weaknesses:**

1. While the paper provides a detailed description of the datasets in terms of their composition and the rules governing the matrices, the process of generating the datasets could be more transparent. The use of a data generator is mentioned, but the specific algorithms and parameters used in the generation process are not described in sufficient detail. This makes it difficult for other researchers to fully understand the datasets.

2. The motivation behind certain design choices in the datasets, such as the specific combinations of held-out attributes and rules in Attributeless-I-RAVEN, could be further elaborated. A more in-depth explanation of how these choices were made and what they are intended to test would enhance the understanding of the datasets' significance.

3. While the paper shows that current models struggle on the proposed benchmarks, the analysis of why generalization is difficult could be deeper. For example, what specific aspects of the held-out attributes or incremental structures make the tasks challenging? Are there any patterns or trends in the model failures that could provide more insights for future improvements?

4. The paper could discuss more about the motivation of design choices of PoNG and why they are effective. For example, what is the intuition behind the parallel pathways and how do they contribute to better generalization?

5. The evaluation of PoNG is mainly limited to RPM benchmarks and visual analogy datasets. It would be beneficial to test the model on a more diverse set of tasks to further validate its generality. As mentioned in the paper, there are other tasks in the AVR domain that could be explored. Moreover, the connection between the proposed benchmarks and real-world applications could be strengthened. How could the results of this research be applied in practical scenarios?

6. I found that the baseline results reported in this paper are inconsistent with several original models. For example, PredRNet and Relbase got 96.5 and 90.5 accuracy on I-RAVEN in their original study. This study only reports them as 88.8 and 89.6. This is also the case for CoPINet and SRAN. I suspect the authors deliberately tuning parameters to get bad results... which is strange and unacceptable.

**Questions:**

I outlined my questions above

---

> ### Author Response · Authors · 2024-11-28
> **Response to Reviewer 7CrM (1/ )**
>
> > While the paper provides a detailed description of the datasets in terms of their composition and the rules governing the matrices, the process of generating the datasets could be more transparent. The use of a data generator is mentioned, but the specific algorithms and parameters used in the generation process are not described in sufficient detail. This makes it difficult for other researchers to fully understand the datasets.
>
> The introduced datasets build upon I-RAVEN, a well-established open-source dataset in the field. The A-I-RAVEN regimes apply constraints on the available choices of attributes and rules between training and test splits, as outlined in Section 3.1. I-RAVEN-Mesh introduces the mesh grid component, detailed in Section 3.2, with Table 1 illustrating the effect of specific rule–attribute pairs applied to the mesh component. We believe that the paper provides a sufficient overview of all important aspects of the introduced datasets, however, if any relevant information is missing we would be grateful for further clarification from the Reviewer. To further support reproducibility, the supplementary material includes the full source code for generating the proposed I-RAVEN variants. The README file provides step-by-step instructions and commands needed to recreate the datasets. If there are specific aspects that require further clarification, we would be happy to address them.
>
> > The motivation behind certain design choices in the datasets, such as the specific combinations of held-out attributes and rules in Attributeless-I-RAVEN, could be further elaborated. A more in-depth explanation of how these choices were made and what they are intended to test would enhance the understanding of the datasets' significance.
>
> The design choices in Attributeless-I-RAVEN were guided by the goal of systematically testing generalization across different levels of complexity. The primary regimes (A/Color, A/Position, A/Size, A/Type) focus on simpler scenarios, allowing for the independent evaluation of generalization to individual attributes. The first three extended regimes (A/ColorSize, A/ColorType, A/SizeType) introduce additional complexity by testing generalization to attribute pairs. Finally, the other three extended regimes (A/Color-Progression, A/Color-Arithmetic, A/Color-DistributeThree) specifically target generalization to the Color attribute with rules other than Constant governing the attribute in the training sets. Together, these regimes span a range of difficulty levels, enabling a more fine-grained analysis of abstract reasoning and generalization capabilities.
>
> > While the paper shows that current models struggle on the proposed benchmarks, the analysis of why generalization is difficult could be deeper. For example, what specific aspects of the held-out attributes or incremental structures make the tasks challenging? Are there any patterns or trends in the model failures that could provide more insights for future improvements?
>
> The A-I-RAVEN regimes are designed to evaluate whether models can form abstract representations of rules and apply them in novel contexts, i.e., to held-out attributes. One notable finding is the weaker performance on A/Type compared to other primary regimes, suggesting that applying rules to object type is particularly difficult. This difficulty may arise from the categorical nature of object type, which differs fundamentally from continuous attributes such as color or size. Categorical attributes often require models to generalize across distinct categories, rather than interpolate between values, which may require better capabilities of forming abstract representations. This observation is further supported by the relatively weaker performance in the A/ColorType and A/SizeType regimes compared to other extended regimes. However, we believe that the main aim of the developed datasets is to serve as effective benchmarks for studying the mechanisms behind knowledge transfer and generalization. Developing components that excel in these settings could help uncover more widely applicable neural network modules that may generalize to other problems. In particular, PoNG demonstrates that models performing well on these benchmarks tend to achieve strong results on other tasks as well, also in the real-world domain.

---

> ### Author Response · Authors · 2024-11-28
> **Response to Reviewer 7CrM (2/ )**
>
> > The paper could discuss more about the motivation of design choices of PoNG and why they are effective. For example, what is the intuition behind the parallel pathways and how do they contribute to better generalization?
>
> Thank you for bringing our attention to this important point. RPMs are inherently hierarchical in nature. Humans typically approach solving RPMs by first identifying the rules governing objects and their attributes in one row, then verifying these rules against another row, and finally selecting an answer that maintains consistency with the discovered rules across all rows. Inspired by this cognitive strategy, we designed the Pathways block to enable the model to discover patterns spanning the entire matrix, as well as specific to certain groups (e.g., rows or pairs of rows).
>
> Consider the first Pathways block, which operates on panel embeddings obtained through the panel encoder. The group convolution module (P3) processes groups of panel embeddings within each row, introducing a row-based inductive bias that facilitates the discovery of row-specific patterns. The group-pair convolution module (P4) processes pairs of row embeddings, enabling the model to identify patterns spanning two rows. P1 and P2 focus on specific spatial locations across all panels, mainly differing in their receptive fields. Although embeddings passed to subsequent Pathways blocks no longer correspond to specific rows or row pairs, they continue to follow the underlying idea of grouping and processing embeddings through shared layers. Additionally, we incorporate TCN after the convolution layers in P3 and P4, following findings from Webb et al. (2020), which demonstrate that TCN enhances generalization when applied to representations in a task-specific window.
>
> > The evaluation of PoNG is mainly limited to RPM benchmarks and visual analogy datasets. It would be beneficial to test the model on a more diverse set of tasks to further validate its generality. As mentioned in the paper, there are other tasks in the AVR domain that could be explored. Moreover, the connection between the proposed benchmarks and real-world applications could be strengthened. How could the results of this research be applied in practical scenarios?
>
> In this work, we evaluated PoNG and 13 reference models (including DRNet and MRNet, which were introduced during the rebuttal) on I-RAVEN, I-RAVEN-Mesh, and 10 A-I-RAVEN regimes, resulting in a total of 12 datasets. **Additionally, PoNG was tested on all 8 PGM regimes, 3 VAP regimes, and 2 VASR splits. This evaluation covers 25 settings and includes a wide range of abstract reasoning problems, including both synthetic and real-world images.**
>
> Our experiments demonstrate that PoNG not only excels on the introduced benchmarks but also achieves strong performance on other RPM datasets such as PGM, and other AVR tasks, including 3 VAP regimes and 2 VASR splits. These tasks cover a diverse range of AVR problems and extend to both synthetic and real-world images.
>
> As described in the “Pathways” paragraph in Section 3, the Pathways block is designed to take an input tensor of shape (B, C, D)---representing a batch where each element is a sequence of vector embeddings—and return an output tensor of the same shape. Given this general formulation, Pathways can be naturally applied to tasks beyond AVR. For instance, the component’s ability to identify patterns across groups makes it particularly well-suited for tasks requiring compositionality or relational reasoning.

---

> ### Author Response · Authors · 2024-11-28
> **Response to Reviewer 7CrM (3/3)**
>
> > I found that the baseline results reported in this paper are inconsistent with several original models. For example, PredRNet and Relbase got 96.5 and 90.5 accuracy on I-RAVEN in their original study. This study only reports them as 88.8 and 89.6. This is also the case for CoPINet and SRAN. I suspect the authors deliberately tuning parameters to get bad results... which is strange and unacceptable.
>
> We appreciate the Reviewer’s observation regarding I-RAVEN results. As we introduced new datasets in our study, we evaluated all considered models on these datasets following a common experimental setup outlined in the “Experimental setup” paragraph in Section 4. In particular, this approach was also applied to the evaluation on I-RAVEN. The noted differences likely arise from differences in hyperparameters between our evaluation and those in the original papers. To improve clarity and facilitate fair comparison, we have revised Table 2 to include the “I-RVN†” column reporting results as stated in the original papers. The revised table shows that with hyperparameters recommended by the authors of the respective models, the best results are achieved by CPCNet (98.5), followed by DRNet (97.6), PredRNet (96.5), and PoNG (95.9). Additionally, in Tables 4, 5, and 6, we explicitly compare PoNG’s performance to the reference results reported in the corresponding original works. We hope this revision addresses the reviewer’s concern and provides a clearer perspective on our evaluation methodology.

---

### Official Review · Reviewer_Lib9 · 2024-11-04

**Soundness:** 3
**Presentation:** 3
**Contribution:** 2
**Rating:** 5
**Confidence:** 3

**Summary:**

The paper proposes two new benchmarks for abstract visual reasoning (AVR) based on Raven’s Progressive Matrices (RPM), called Attributeless-I-RAVEN comprising 10 generalization regimes, and I-RAVEN-Mesh, an extension of I-RAVEN with grid lines overlaid to evaluate progressive generalization in transfer learning setting. The authors evaluate the performance of state-of-the-art AVR models on these benchmarks, demonstrating their inability to generalize, and develop a new AVR model, called PoNG. PoNG consists of an encoder which generates an embedding for each image panel constituting the RPM problem, a reasoner module that processes the panel embeddings, and prediction heads for predicting the answer and the RPM rules. The reasoner module has four parallel pathways implementing standard 1D convolutions, grouped, and group-paired convolutions. The authors show that PoNG demonstrates state-of-the-art performance on the two proposed benchmarks, alongwith improvements on previous AVR benchmarks, I-RAVEN, PGM and visual analogy problems involving real-world images. Ablation studies demonstrate the effectiveness of grouped and group-paired convolutions, normalization in the parallel pathways of the reasoner module, and auxiliary losses for predicting matrix rules.

**Strengths:**

1. The authors propose two AVR benchmarks, Attributeless-I-RAVEN and I-RAVEN-Mesh for measuring the generalization and transfer learning capabilities of AVR models.

2. The authors also propose a new AVR model, Pathways of normalized group convolution (PoNG), which demonstrate state of the art performance on the two proposed benchmarks.

3. PoNG also demonstrates improvements on previous AVR benchmarks, I-RAVEN, PGM and visual analogy problems involving real world images.

4. The paper is well-written and easy to follow.

**Weaknesses:**

1. The performance of the proposed model PoNG seems to heavily depend on auxiliary losses for predicting matrix rules, which might not be available or easily obtained for RPM datasets. If the other baseline methods aren’t trained with the auxiliary losses, the comparison is unfair.

2. The motivation behind the design of parallel pathways in the reasoner module is missing, and it is also not clear how this can be generally useful for non-AVR settings.

**Questions:**

1. How are the values of loss weights $\beta$ and $\gamma$ in line 340 chosen?

2. Are the baseline models also trained with the auxiliary loss to predict the rules like PoNG?

3. It would be informative to see which of the two pathways (P3 or P4) play a major role in PoNG’s performance? Also are 4 four parallel pathways really necessary or just P3 and P4 are enough?

4. For I-RAVEN, and the two proposed benchmarks the authors should also evaluate ARII [1].

5. For PGM, the authors should also compare to other strong-performing baseline methods like ARII [1], MRNet [2], and Slot-Abstractors [3].

6. Can the authors share some intuition behind PoNG performing so well for Heldout Attribute pairs, and Heldout Triple pairs regimes of the PGM dataset but struggle with regimes like H-O-L-T, H-O-S-C, and Extrapolation?

7. The authors should also report the average performance over all PGM regimes in Table 4,  to better understand the overall SOTA performance.

8. A more rigorous test of transfer learning would be to test zeroshot performance on I-RAVEN-Mesh, with models only pretrained on I-RAVEN.

[1] -  Zhang, W., Mo, S., Liu, X. and Song, S., 2022. Learning robust rule representations for abstract reasoning via internal inferences. Advances in Neural Information Processing Systems, 35, pp.33550-33562.

[2] - Benny, Y., Pekar, N. and Wolf, L., 2021. Scale-localized abstract reasoning. In Proceedings of the IEEE/CVF Conference on Computer Vision and Pattern Recognition (pp. 12557-12565).


[3] -  Mondal, S.S., Cohen, J.D. and Webb, T.W., 2024. Slot abstractors: Toward scalable abstract visual reasoning. arXiv preprint arXiv:2403.03458.

---

> ### Author Response · Authors · 2024-11-28
> **Response to Reviewer Lib9 (1/ )**
>
> > The performance of the proposed model PoNG seems to heavily depend on auxiliary losses for predicting matrix rules, which might not be available or easily obtained for RPM datasets. If the other baseline methods aren’t trained with the auxiliary losses, the comparison is unfair.
> Are the baseline models also trained with the auxiliary loss to predict the rules like PoNG?
>
> Thank you for raising this important concern. We fully agree that a fair comparison requires training all models with auxiliary losses, and this was indeed the case in our experiments. Specifically, all baseline models were trained using both the “target head” and “aggregate rule head”, as described in the “Answer prediction” paragraph in Section 3. The “target-conditioned rule head”, however, is a novel component introduced in our work and was employed exclusively for training PoNG. We have clarified this experimental setup in the revision under the “Experimental setup” paragraph in Section 4.
>
> > The motivation behind the design of parallel pathways in the reasoner module is missing, and it is also not clear how this can be generally useful for non-AVR settings.
>
> Thank you for bringing our attention to this important point. RPMs are inherently hierarchical in nature. Humans typically approach solving RPMs by first identifying the rules governing objects and their attributes in one row, then verifying these rules against another row, and finally selecting an answer that maintains consistency with the discovered rules across all rows. Inspired by this cognitive strategy, we designed the Pathways block to enable the model to discover patterns spanning the entire matrix, as well as patterns specific to certain groups (e.g., rows or pairs of rows).
>
> Consider the first Pathways block, which operates on panel embeddings obtained through the panel encoder. The group convolution module (P3) processes groups of panel embeddings within each row, introducing a row-based inductive bias that facilitates the discovery of row-specific patterns. The group-pair convolution module (P4) processes pairs of row embeddings, enabling the model to identify patterns spanning two rows. P1 and P2 focus on specific spatial locations across all panels, mainly differing in their receptive fields. Although embeddings passed to subsequent Pathways blocks no longer correspond to specific rows or row pairs, they continue to follow the underlying idea of grouping and processing embeddings through shared layers. Additionally, we incorporate TCN after the convolution layers in P3 and P4, following findings from Webb et al. (2020), which demonstrate that TCN enhances generalization when applied to representations in a task-specific window.
>
> Our experiments demonstrate that the Pathways block not only excels on the introduced benchmarks but also achieves strong performance on other RPM datasets such as PGM, and other AVR tasks, including three VAP regimes and two VASR splits. These tasks cover a diverse range of AVR problems and extend to both synthetic and real-world images. As described in the “Pathways” paragraph in Section 3, the Pathways block is designed to take an input tensor of shape (B, C, D)---representing a batch where each element is a sequence of vector embeddings—and return an output tensor of the same shape. Given this general formulation, Pathways can be naturally applied to tasks beyond AVR. For instance, the component’s ability to identify patterns across groups makes it particularly well-suited for tasks requiring compositionality or relational reasoning.
>
> > How are the values of loss weights beta and gamma in line 340 chosen?
>
> During model development, we manually optimized several hyperparameters, including $\beta$ and $\gamma$, as well as others detailed in Appendix D. For $\beta$ and $\gamma$, we explored values from the set {$1, 5, 10, 25$}. Due to computational constraints, we did not perform a comprehensive grid search but instead evaluated a limited set of manually selected configurations. Model performance was assessed on I-RAVEN, I-RAVEN-Mesh, and A/Color, and the configuration yielding the best results was selected. It is important to note that the hyperparameters were not tuned for the remaining 9 A-I-RAVEN regimes or for other datasets used in the paper. In the “Ablation study” paragraph in Section 4, we present a more comprehensive evaluation of model performance to better understand the impact of these design choices on additional A-I-RAVEN regimes.

---

> ### Author Response · Authors · 2024-11-28
> **Response to Reviewer Lib9 (2/ )**
>
> > For I-RAVEN, and the two proposed benchmarks the authors should also evaluate ARII [1].
> > For PGM, the authors should also compare to other strong-performing baseline methods like ARII [1], MRNet [2], and Slot-Abstractors [3].
>
> Thank you for bringing these relevant works to our attention. During the rebuttal period, we conducted further experiments on the introduced datasets. Due to time constraints, we focused on two recent models mentioned by the Reviewers, including DRNet and MRNet. In summary, on I-RAVEN DRNet reports 97.6% accuracy, outperforming PoNG (95.9%) by 1.7 p.p. Under the experimental setup employed in our work, DRNet demonstrates strong performance, securing the 2nd best results on I-RAVEN, A/Color, A/Position (Table 2), and A/Color-Arithmetic (Table 10). Conversely, MRNet delivers competitive performance, but underperforms relative to other reference models. Nevertheless, PoNG outcompetes DRNet and MRNet on I-RAVEN-Mesh and all A-I-RAVEN regimes. Experments regarding progressive knowledge acquisition on I-RAVEN-Mesh show that DRNet presents strong transfer learning capacity (Fig. 6).
>
> In addition, we have included PGM results for MRNet, ARII, DRNet, and Slot-Abstractor in Table 4. On average, DRNet achieves the highest accuracy (58.3%), surpassing PoNG (57.3%) by 1 p.p. PoNG secures the 2nd place in the Neutral, Held-out Attribute Pairs, and Held-out Triples regimes, while achieving the 1st place in the Held-out Triple Pairs regime.
>
> We hope that these extended results provide additional context for positioning PoNG against state-of-the-art models. Furthermore, the evaluation of DRNet and MRNet on I-RAVEN-Mesh and A-I-RAVEN extends the scope of baseline results on our introduced datasets, facilitating future comparisons by providing a more comprehensive reference.
>
> > It would be informative to see which of the two pathways (P3 or P4) play a major role in PoNG’s performance? Also are 4 four parallel pathways really necessary or just P3 and P4 are enough?
>
> Thank you for this suggestion. To better understand the contributions of individual pathways, we conducted additional experiments with PoNG excluding the P1 and P2 pathways. The modified model still achieves reasonable performance on I-RAVEN (92.8%, -3.1 p.p. compared to the main configuration) and A/Position (76.4%, −2.9 p.p.), indicating that P3 and P4 alone can partially compensate for the absence of P1 and P2. However, relatively weaker results in the remaining setups emphasize that PoNG achieves best performance when all four pathways are included. We have added these results as “w/o P1 and P2” to Table 2.
>
> Additionally, we have initiated further experiments to evaluate PoNG w/o P3 and w/o P4 to better isolate the contributions of the respective pathways. We will report on these results once the experiments are completed.

---

> ### Author Response · Authors · 2024-11-28
> **Response to Reviewer Lib9 (3/3)**
>
> > Can the authors share some intuition behind PoNG performing so well for Heldout Attribute pairs, and Heldout Triple pairs regimes of the PGM dataset but struggle with regimes like H-O-L-T, H-O-S-C, and Extrapolation?
>
> Thank you for raising this question. Understanding why PoNG performs well on certain PGM regimes (e.g., Held-out Attribute Pairs and Held-out Triple Pairs) but struggles with others (e.g., Held-out Line-Type, Held-out Shape-Colour, and Extrapolation) requires careful consideration of the nature of these regimes.
>
> In HO-AP and HO-TP regimes, the model encounters novel combinations of attributes or triples during testing but still benefits from having seen other configurations involving the same objects, attributes and rules during training. For instance, in HO-AP, while a specific attribute pair may be held out, the model has likely observed various rules applied to each attribute in other contexts, enabling it to form generalizable representations. Similarly, in HO-TP, the model learns from the training set how to process individual triples, enabling it to generalize to unseen triple pairs. These regimes are closely related to interpolation, a setting where neural networks are known to excel compared to extrapolation.
>
> The HO-LT and HO-SC regimes define held-out attribute pairs (i.e., line type and shape-color) that are absent in the training set. The model does not see any rules involving these object-attribute pairs during training and may therefore fail to build meaningful representations of them, assuming they’re irrelevant to minimize the training objective. Similarly, in the Extrapolation regime, the model must generalize to attribute values outside the range encountered during training. These 3 regimes are closely related to extrapolation, a known challenge for neural networks, as evidenced by overally weaker performance on HO-LT, HO-SC, and Extrapolation regimes across all models in prior studies.
>
> While this analysis provides an intuition behind PoNG’s performance, a deeper analysis would require ablation studies to isolate the impact of model components on each regime. However, conducting such studies on PGM is computationally expensive due to the dataset’s large size. To address this limitation, we developed smaller benchmarks, including I-RAVEN-Mesh and A-I-RAVEN, that enable fine-grained analysis of model generalization capabilities. As exemplified by PoNG, advancements on these smaller datasets transfer not only to larger RPM benchmarks such as PGM but also to other AVR tasks.
>
> > The authors should also report the average performance over all PGM regimes in Table 4, to better understand the overall SOTA performance.
>
> Thank you for this suggestion, we have added the “Avg.” column to Table 4 to report the average performance across all PGM regimes.
>
> > A more rigorous test of transfer learning would be to test zeroshot performance on I-RAVEN-Mesh, with models only pretrained on I-RAVEN.
>
> Thank you for this valuable suggestion. We agree that such an evaluation could provide additional insights regarding zero-shot generalization capabilities of the tested models. We have initiated the suggested experiments and will report the results in the coming days.

---

> > ### Comment · Reviewer_Lib9 · 2024-12-01
> > **Official Comment by Reviewer Lib9**
> >
> > I thank the authors for responding to my questions and incorporating my feedback. In table 2 (column I-RVN) the authors compare to results on I-RAVEN as reported in the original papers, where models aren't trained with auxiliary losses, but PoNG result of 95.9 uses auxiliary losses. The correct comparison should be with $\gamma =0, \beta=0$ where PoNG achieves 79.7. This suggests that PoNG heavily relies on auxiliary losses, unlike other models. Also for PGM results in Table 4, the baseline models aren't trained with auxiliary loss, but if PoNG is, then it is an unfair comparison. In light of these issues, I will stick with my rating.

---

> > > ### Author Response · Authors · 2024-12-03
> > > **Response to Reviewer Lib9**
> > >
> > > We thank the Reviewer for their feedback and acknowledge that the use of auxiliary loss is an important aspect to account for in evaluating model performance. PoNG’s reliance on such auxiliary information was explicitly noted in the ablation study: *"[..] signifying high relevance of the auxiliary training signal in PoNG’s training."*. We also utilized auxiliary loss to evaluate the model on PGM. However, we’d like to emphasize that PoNG represents only one out of four contributions outlined in Section 1. In addition to PoNG:
> > > 1. We introduce the Attributeless-I-RAVEN (A-I-RAVEN) dataset that enables measuring generalization across 10 regimes.
> > > 1. We construct I-RAVEN-Mesh, an extension of I-RAVEN with a new component structure that facilitates assessment of progressive knowledge acquisition in a TL setting.
> > > 1. We evaluate the performance of state-of-the-art AVR models on the introduced benchmarks, uncovering their limitations in terms of generalization to novel problem settings.
> > >
> > > We hope the above-mentioned contributions constitute an important addition to existing state-of-the-art research in the field of AVR and are valuable, even if the Reviewer does not rate PoNG as a major achievement.

---

### Official Review · Reviewer_CjoB · 2024-11-04

**Soundness:** 2
**Presentation:** 3
**Contribution:** 3
**Rating:** 5
**Confidence:** 4

**Summary:**

This paper investigates the generalization and knowledge transfer capabilities of DNNs in abstract visual reasoning tasks. The authors introduce two new datasets, A-I-RAVEN and I-RAVEN-Mesh to assess generalization capacity of reasoning and progressive knowledge acquisition in transfer learning setting, respectively.  They show the lack of generalization for the contemporary models and propose PoNG to solve the problems.  The results reveals improvement over SOTA reference models.

**Strengths:**

1. Two new dataset were introduced to evaluate the generatilization and transfer learning in abstract visual reasoning tasks.

2. A new model PoNG was proposed to improve the generalization capacity of abstract visual reasoning.

3. PoNG aslo performs well on other AVR tasks (visual analogy problem),  showcasing its generality in analogical reasoning.

**Weaknesses:**

One main problem:

The I-RAVEN results from Table 2 are different from those in the original paper.

**Questions:**

1. The I-RAVEN results from Table 2 are different from those in the original paper. For example, the result of PredRNet in original paper [1] is 96.5 while the authors report it as 88.8.

2. Some SOTA models are not included in the paper, such as DRNet [2] and SLOT-ABSTRACTOR [3].



[1]Yang L, You H, Zhen Z, et al. Neural prediction errors enable analogical visual reasoning in human standard intelligence tests[C]//ICML, 2023: 39572-39583.

[2]Zhao K, Xu C, Si B. Learning Visual Abstract Reasoning through Dual-Stream Networks[C]//AAAI. 2024, 38(15): 16979-16988.

[3]Mondal S S, Cohen J D, Webb T W. Slot abstractors: Toward scalable abstract visual reasoning[C]//ICML, 2024.

---

> ### Author Response · Authors · 2024-11-28
> **Response to Reviewer CjoB**
>
> > The I-RAVEN results from Table 2 are different from those in the original paper. For example, the result of PredRNet in original paper [1] is 96.5 while the authors report it as 88.8.
>
> We appreciate the Reviewer’s observation regarding I-RAVEN results. As we introduced new datasets in our study, we evaluated all considered models on these datasets following a common experimental setup outlined in the “Experimental setup” paragraph in Section 4. In particular, this approach was also applied to evaluations performed on I-RAVEN. The noted differences most likely arise from the differences in hyperparameters between our evaluation and those in the original papers. To improve clarity and facilitate a fair comparison, we have revised Table 2 to include the “I-RVN†” column reporting results as stated in the original papers. The revised table shows that with hyperparameters recommended by the authors of the respective models, the best results are achieved by CPCNet (98.5), followed by DRNet (97.6), PredRNet (96.5), and PoNG (95.9). Additionally, in Tables 4, 5, and 6, we explicitly compare PoNG’s performance to the reference results reported in the corresponding original works. We hope this revision addresses the reviewer’s concern and provides a clearer perspective on our evaluation methodology.
>
> > Some SOTA models are not included in the paper, such as DRNet [2] and SLOT-ABSTRACTOR [3].
>
> Thank you for bringing these recent relevant works to our attention. Based on your suggestion, during the rebuttal period we conducted additional experiments on the introduced datasets. Due to time constraints, we focused on two recent models mentioned by the Reviewers, including DRNet and MRNet. In summary, on I-RAVEN DRNet reports 97.6% accuracy, outperforming PoNG (95.9%) by 1.7 p.p. With the experimental setup employed in our work, DRNet demonstrates strong performance, securing the 2nd best results on I-RAVEN, A/Color, A/Position (Table 2), and A/Color-Arithmetic (Table 10). Conversely, MRNet delivers competitive performance, but underperforms relative to other reference models. Nevertheless, PoNG outcompetes DRNet and MRNet on I-RAVEN-Mesh and all A-I-RAVEN regimes. Experments regarding progressive knowledge acquisition on I-RAVEN-Mesh show that DRNet presents strong transfer learning capacity (Fig. 6).
>
> In addition, we have included PGM results for MRNet, ARII, DRNet, and Slot-Abstractor in Table 4. On average, DRNet achieves the highest accuracy (58.3%), surpassing PoNG (57.3%) by 1 p.p. PoNG secures the 2nd place in the Neutral, Held-out Attribute Pairs, and Held-out Triples regimes, while achieving the 1st place in the Held-out Triple Pairs regime.
>
> We hope that these extended results provide additional context for positioning PoNG against state-of-the-art models. Furthermore, the evaluation of DRNet and MRNet on I-RAVEN-Mesh and A-I-RAVEN extends the scope of baseline results on our introduced datasets, facilitating future comparisons by providing a more comprehensive reference.

---

### Note · Authors · 2025-01-19

I have read and agree with the venue's withdrawal policy on behalf of myself and my co-authors.